



# The Vulcan Version 3.0 High-Resolution Fossil Fuel CO$_2$ Emissions for the United States

Kevin R. Gurney[1], Jianming Liang[2], Risa Patarasuk[2], Yang Song[2], Jianhua Huang[2], Geoffrey Roest[1]

[1]School of Informatics, Computing, and Cyber Systems, Northern Arizona University, Flagstaff, AZ, USA
[2]School of Life Sciences, Arizona State University, Tempe AZ USA

*Correspondence to*: Kevin R. Gurney (kevin.gurney@nau.edu)

**Abstract**. Estimates of greenhouse gas emissions, quantified at fine space and time scales, has become a critical component of new multi-constraint flux information systems in addition to providing relevant information to decisionmakers when considering GHG mitigation opportunities. The 'Vulcan Project' is an effort to estimate bottom-up fossil fuel emissions and CO$_2$ emissions from cement production (FFCO$_2$) for the entire United States landscape at space and time scales that satisfy both scientific and policy needs. Here, we report on version 3.0 of the Vulcan emissions which quantifies FFCO$_2$ emissions for the U.S. at a spatial resolution of 1km x 1km and hourly temporal resolution for the 2010-2015 time period. We provide a complete description of the updated methods, data sources, results, and comparison to a global gridded FFCO$_2$ data product. We estimate FFCO$_2$ emissions for the year 2011 of 1589.3 TgC with a 95% confidence interval of 1299/1917 TgC (+18.3%/-20.6%), implying a one-sigma uncertainty of ~ ±10%. We find that per capita FFCO$_2$ emissions are larger in states dominated by the electricity production and industrial sectors and smaller in states dominated by onroad and residential/commercial building emissions. The center of mass (CoM) of FFCO$_2$ emissions in the US are located in the state of Missouri with mean seasonality that moves on a NE/SW near-elliptical path. Comparison to ODIAC, a global gridded FFCO$_2$ emissions estimate shows large differences in both total emissions (100.1 TgC for year 2011) and spatial patterns. The spatial correlation (R$^2$) between the two data products was 0.38 and the mean absolute difference at the individual gridcell scale was 80.04%. The Vulcan v3.0 FFCO$_2$ emissions data product offers an immediate high-resolution estimate of emissions in every city within the U.S., providing a large potential savings of time and effort for cities planning to develop self-reported city inventories. The Vulcan v3.0 annual gridded emissions data product can be downloaded from the Oak Ridge National Laboratory Distributed Active Archive Center (ORNL DAAC) (https://doi.org/10.3334/ORNLDAAC/1741, Gurney et al., 2019).

## 1 Introduction

Global emissions of carbon dioxide from the combustion of fossil fuels (FFCO$_2$) comprise the largest net flux of carbon into the Earth's atmosphere and remain the primary driver of anthropogenic climate change (IPCC 2013; USGCRP 2018). Improving our quantitative understanding of FFCO$_2$ fluxes remains a critical component of climate change research and climate policy. For example, scientific understanding of the global carbon cycle and how it interacts with climate change rests on accurate quantification of FFCO$_2$ emissions at multiple scales (LeQuere,



2018). This, in turn, improves the reliability of future projections of climate change and specifies the emissions
reductions necessary to meet specific targets, such as limiting the rise of global mean temperature to 1.5 C (IPCC
2018). Understanding $FFCO_2$ sources also assists in understanding the composition, driving factors, and
responsibility for emissions, making mitigation options better-targeted, equitable and ultimately more effective
(Durant et al., 2011; Janssens-Maenhout et al., 2013; Bellassen et al., 2015).
Quantification of $FFCO_2$ emissions began as efforts to capture total emissions at the global and national spatial
scales aiming to quantify anthropogenic fluxes to better understand the drivers of climate change and the global
carbon cycle. Employing accounting approaches that rely on national statistics of energy production and
consumption, a number of national and international institutions produce and archive estimates of $FFCO_2$ emissions,
often disaggregated to economic sector and fuel type (see reviews by Andres et al., 2012; Macknick, 2011). In
response to the advances in carbon cycle observations and modeling studies, many of these $FFCO_2$ inventory
products began to increase their spatial and temporal resolution below the nation-state, often representing emissions
in regularized gridded format (Marland et al., 1985; Andres et al., 1996; Olivier et al., 1999). Gridded output was
especially important when used within systems that solve carbon fluxes through inversion of atmospheric transport
constrained by atmospheric concentration measurements (Gurney et al., 2002; 2005; Peylin et al., 2011; Liu et al.,
2014; Yadav et al., 2016; Gaubert et al., 2019). Most often these sub-national representations of $FFCO_2$ emissions
used proxy information, such as population statistics or remotely-sensed nighttime lights, to distribute the
national/global emissions to smaller space/time scales (Andres et al., 1999, Olivier et al., 2005; Rayner et al., 2010;
Ghosh et al., 2010, Oda and Maksyutov, 2011; Ou et al., 2015). Recent research has employed a mixture of global
"bottom-up" information such as powerplant databases with remote-sensing information (Wang et al., 2013; Oda et
al., 2018  etc), sometimes within optimization frameworks to more mechanistically distribute emissions in space and
time in addition to offering more formal uncertainty estimation (Asefi et al., 2014).
In addition to the globally gridded representations, research effort has also aimed at specific national and regional
domains often with additional detail on the emitting process (Gregg and Andres, 2007; Gregg et al., 2009; Bun et al.,
2007; 2018; Gately et al., 2017; Kurokawa et al., 2013; Ivanova et al., 2017; Denier et al., 2017; Cai et al., 2018)
with some studies focussed on an individual sector or source type (Petron et al., 2008; Gately et al., 2013; Zheng et
al., 2014; Wang et al., 2014; Liu et al., 2015). These national and regional efforts were often modeled after work in
local air pollution inventories (Cooke et al., 1999; Baldesano et al., 2008; O'Hara et al., 2007; Hoesly et al., 2017).
In addition to focusing on different national domains with unique datasets, many of these past research efforts reflect
different methodological approaches to data interpretation, downscaling and modeling. Many of these efforts,
however, follow the general approach established in the pioneering work of the Vulcan Project, the first attempt to
generate a completely bottom-up space/time-explicit national estimate of all $FFCO_2$ emission sources (Gurney et al.,
2009). The Vulcan Project, which estimated $FFCO_2$ emissions at the "native" resolution of emission points, lines,
and polygons, produced US $FFCO_2$ emissions on a 10km x 10km spatial grid at hourly time resolution for the year
2002. Used in a variety of research and applied policy settings, the Vulcan Project has spawned additional efforts at
downscaling into the urban domain, where resolution has gone to the scale of individual buildings and street



segments for whole urban areas (VandeWeghe and Kennedy 2007; Shu and Lam 2011; Zhou and Gurney 2011;
Gurney et al., 2012; Wilson et al., 2013; Pincetl et al., 2014; Patarasuk et al. 2016; Gurney et al., 2018; 2019b).
All of the FFCO$_2$ emissions reviewed thus far are categorized as "scope 1" or "in-boundary" emissions. They are an
accounting of emissions that reflects physical emission of CO$_2$ molecules from the geography resolved (e.g. gridcell,
state, province). This is in contrast to quantification of fluxes that assign emissions to consumptive activity such as
using electricity or consuming food (Davis and Caldeira, 2010). The two accounting perspectives are identical at the
whole-planet scale but diverge as one considers scales at the nation-state or below. Consumption-based FFCO$_2$
emissions quantification has a long history at scales ranging from the nation-state to the city, but has only recently
begun to systematically resolve (e.g. in gridded form) FFCO$_2$ emissions below the nation-state scale (Jones and
Kammen, 2011; 2013; Zhang et al., 2014; Minx et al., 2013; Moran et al., 2018). The current study emphasizes in-
boundary emissions because these can be directly used with atmospheric monitoring, a critical element in
evaluation/validating the estimated fluxes and a motivation for the research reported here (NRC 2010).
In this paper, we introduce a significant update to the Vulcan Project estimation of high-resolution US fossil fuel
carbon dioxide emissions and CO$_2$ emissions from cement production (collectively referred to here as "FFCO$_2$").
We report here on improvements in methodology, resolution, uncertainty estimation, in addition to more
contemporaneous, multiyear output. We present some of the fundamental results of the Vulcan output and compare
to the only commensurate resolved FFCO$_2$ emissions data product covering the entire U.S. landscape, the ODIAC
global data product. We show results associated with a few zoomed urban locations, suggesting that the Vulcan
FFCO$_2$ emissions data product has a role to play in providing U.S. cities with a sub-city resolved scope 1 CO$_2$
emissions inventory.
Version 3.0 of the Vulcan data product and associated documentation is publicly available and annual, gridded,
multiyear results can be downloaded from the Oak Ridge National Laboratory Distributed Active Archive Center
(ORNL DAAC) (https://doi.org/10.3334/ORNLDAAC/1741).
This paper is structured as follows: In section 2, we describe the data and model processes used to generate the
Vulcan version 3.0 (v3.0) FFCO$_2$ emissions data product including those used for spatial and temporal distribution.
In section 3.0, we present the results, the uncertainties and a series of descriptive statistics at various scales of
aggregation. In section 4, we compare Vulcan to the ODIAC data product, and discuss the potential use and
relevance of this work, known gaps and weaknesses, in addition to next steps and future work.
**2 Methods**
The Vulcan version 3.0 FFCO$_2$ emissions data product represents total FFCO$_2$ emissions resulting from the
combustion of fossil fuel (coal, petroleum and natural gas) and the CO$_2$ from cement production in the 50 United
States and District of Columbia for 2010-2015 time period (Gurney et al., 2019). It is constructed from numerous
public datasets that generate emissions magnitude, the spatial representation, and temporal representation of those
emissions. The FFCO$_2$ emissions are initially estimated at their "native" spatial and temporal resolution (e.g.
counties, points, lines, annual, hourly) depending upon the characteristics of the incoming data sources. Additional
spatial and temporal "conditioning" (e.g. downscaling, interpolation, proxy surrogates), where needed, is used to
arrive at an hourly representation for six complete calendar years (2010-2015) at the spatial resolutions of a US
Census block-group or finer (e.g. points, lines). The $FFCO_2$ emissions are further processed to regularized hourly
grids at a resolution of 1 km x 1 km, for the contiguous United States and Alaska. The $FFCO_2$ emissions represent
all fossil fuel combustion extending 12 nautical miles from the coastal boundary of the United States.
**2.1 Data and processing**
The data sources for the $FFCO_2$ emissions estimation are organized here by data source *type* and/or the economic
*sector* in accordance with original data collection/categorization (see Table 1). This paper describes the scientific
methodology used to generate the Vulcan v3.0 $FFCO_2$ emissions but should be considered in combination with the
published results for the earlier version 2.0 Vulcan results (Gurney et al., 2009; Zhou et al., 2011) and the Vulcan
version 2.0 documentation (http://vulcan.rc.nau.edu/assets/files/Vulcan.documentation.v2.0.online.pdf).
Uncertainty quantification relies on the characterization of a 95% confidence interval (CI). Due to the considerable
runtime of the Vulcan codebase, only the boundaries of the upper and lower CI are estimated (referred to as "hi" and
"lo" CI bounds). Future versions of the Vulcan data product will quantify the complete uncertainty distribution of
the Vulcan $FFCO_2$ emissions output.
**Table 1: Overview of data sources used in generating the space/time-resolved Vulcan v3.0 $FFCO_2$ emissions**
**(footnotes provide acronym explanations).**

| Sector/type | Emissions Data Source | Original spatial resolution/information | Spatial distribution | Temporal distribution |
|---|---|---|---|---|
| Onroad | EMFAC [a] CO2, EPA NEI[b] onroad $CO_2$ | County, road class, vehicle class | FHWA AADT[c] | CCS[e] |
| Electricity production | CAMD[f] CO2, DOE/EIA[g] fuel, EPA NEI point CO | Lat/lon, fuel type, technology | EPA/EIA NEI Lat/Lon, Google Earth | CAMD, EIA and EPA |
| Residential nonpoint buildings | EPA NEI nonpoint CO | County, fuel type | FEMA HAZUS[d], DOE RECS NE-EUI[h] | eQUEST[i] model |
| Nonroad | NEI nonpoint CO | County, vehicle class | EPA spatial surrogates (vehicle class specific) | EPA temporal surrogates (by SCC[j]) |
| Airport | EPA NEI point CO | Lat/lon, aircraft class | Lat/Lon | LAWA & OPSNET[k] |
| Commercial nonpoint buildings | EPA NEI nonpoint CO | County, fuel | FEMA HAZUS, DOE CBECS NE-EUI[l] | eQUEST model |
| Commercial point sources | EPA NEI point CO | Lat/lon, fuel type, combustion technology | EPA NEI Lat/Lon, Google Earth | eQUEST model |
| Industrial point sources | EPA NEI point CO | Lat/Lon, fuel type, combustion technology | EPA NEI Lat/Lon, Google Earth | EPA temporal surrogates (by SCC) |
| Industrial nonpoint buildings | EPA NEI nonpoint CO | County, fuel type | FEMA HAZUS, DOE MECS NE-EUI[m] | eQUEST model |
| Commercial Marine Vessels | EPA NEI nonpoint CO | County, fuel type, port/underway | EPA port and shipping lane shapefiles | Flat time structure |
| Railroad | EPA NEI nonpoint CO, EPA NEI point CO | County, fuel type, segment | EPA NEI rail shapefile and density distribution | Point records: EPA temporal surrogates (by SCC). Nonpoint: flat time structure |
| Cement | Portland Cement Association, USGS | Lat/lon | PCA lat/lon checked in Google Earth | Flat time structure |

a.  Emissions Factors Model
b.  Environmental Protection Agency, National Emissions Inventory
c.  Federal Highway Administration, Annual Average Daily Traffic
d.  Federal Emergency Management Agency



e. Continuous Count Stations
f. Clean Air Markets Division
g. Department of Energy/Energy Information Administration
h. Department of Energy Residential Energy Consumption Survey, non-electric energy use intensity
i. Quick Energy Simulation Tool
j. Source Classification Code
k. Los Angeles World Airport, The Operations Network
l. Department of Energy Commercial Energy Consumption Survey, non-electric energy use intensity
m. Department of Energy Manufacturing Energy Consumption Survey, non-electric energy use intensity

### 2.1.2 Nonpoint sources

The area or nonpoint source emissions (dominated by residential and commercial economic sectoral categories) are stationary sources that are not inventoried at the individual facility-level and can be thought of as representing "diffuse" or dispersed sources within a geographic area. Vulcan nonpoint $FFCO_2$ emissions are estimated using a number of data sources. Foremost among these are the Environmental Protection Agency (EPA) National Emission Inventory (NEI) nonpoint reporting for carbon monoxide (CO) emissions, version 2 for the year 2011 (USEPA 2015a). The NEI is a comprehensive inventory of all criteria air pollutants (CAPs) and hazardous air pollutants (HAPs) across the United States (USEPA 2005a). The NEI now includes greenhouse gases for select sectors (onroad, nonroad). The NEI is a data structure with which the EPA can meet mandates established by the Clean Air Act (CAA). The CAP emissions, the component of emissions used by the Vulcan system (other than onroad, nonroad, and electricity production), are collected under the Air Emissions Reporting Rule (40 CFR Part 51) (CFR, 2008). The NEI can be used to track progress, drive air quality modeling, enable emissions trading, and ensure comprehensive reporting and compliance.

The emissions data within the NEI are collected from state, local, and tribal (SLT) agencies and augmented by numerous federal data sets such as the Toxics Release Inventory (TRI), the Acid Rain Program (ARP), and the Federal Highway Administration (FHWA) traffic counts.

The EPA provides recommendations to SLT agencies on how to collect nonpoint source emissions information and the SLT agencies are given a number of options in forming the basis of the reported information (ERG 2001). The EPA prefers emissions to be estimated by extrapolating from a sample set of data for the activity to the entire population, but a number of other approaches are allowed including material balance, mathematical models, and emission factors (EFs). This means that the method employed will vary by location. The EPA will augment the submitted data as a result of recognized data gaps, QA/QC procedures, or in consultation with SLT agencies.

The 2011 NEIv2 nonpoint data used in the Vulcan emissions estimation is composed of two core data files. These data files share common, required key fields. The fundamental nonpoint "unit", as pertains to the Vulcan calculations, is a reported combustion process emitting carbon monoxide (CO) identified by a single source classification code (SCC) in a single US county burning an identified fossil fuel. The numerical SCC (USEPA, 1995) and FIPS values (which identifies the state and county via numerical ID) are critical common IDs. Reporting associated with fugitive emissions (non-combustion), chemical or "in-process", or resulting from the combustion of biogenic fuel sources are removed. An exception to this is the in-process emissions associated with cement production, however these emissions are generated with different data outlined in a later section. Fuels considered in



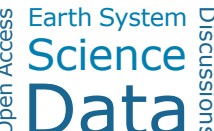

the Vulcan nonpoint FFCO₂ estimation along with their thermodynamic heat value, default CO emission factor (EF),
and CO₂ EF are provided in Table 2.
**Table 2: Heat value, carbon monoxide emission factor, and carbon dioxide emission factor for emission**
**sources. Square brackets denote instances in which different emission factors are used in application to point**
**versus nonpoint data.**

| Sector | Fuel | HV (e6btu/unit) | Unit | CO EF (lbs/e9btu)ᶠ | CO₂ EF (tC/e9btu) |
|---|---|---|---|---|---|
| Electricity Production | Bituminous Coal | 26.50ᶜ | Tonne | 247 | 25.4 |
| Electricity Production | Subbituminous Coal | 19.30ᶜ | Tonne | 344 | 25.9 |
| Electricity Production | Bituminous/Subbituminous Coal | 22.90ᶜ | Tonne | 295 | 25.9 |
| Electricity Production | Coal | 22.90ᶜ | Tonne | 29 | 25.9 |
| Electricity Production | Anthracite | 27.49ᶜ | Tonne | 24 | 28.2 |
| Electricity Production | Lignite | 14.29ᶜ | Tonne | 39 | 26.2 |
| Electricity Production | Natural Gas | 1032.00¶ | e6ft3 | 63 | 14.5 |
| Electricity Production | Distillate Oil | 139.93§ | e3gal | 36 | 19.8 |
| Electricity Production | Residual Oil | 149.97§ | e3gal | 33 | 21.3 |
| Electricity Production | Liquified Petroleum Gas (LPG) | 94.00§ | e3gal | 28 | 16.9 |
| Electricity Production | Process Gas | 1068.57¶ | e6ft3 | 33 | 15.3 |
| Electricity Production | Coke | 30.82§ | tonne | 21 | 27.6 |
| Electricity Production | Distillate Oil (Diesel)/Diesel | 137.06§ | e3gal | 929 | 20.0 |
| Electricity Production | Oil | 138.69¶ | e3gal | 36 | 19.8 |
| Electricity Production | Jet Fuel | 120.19§ | e3gal | 751 | 19.2 |
| Electricity Production | Refinery Gas | 1068.57¶ | e6ft3 | 33 | 15.3 |
| Industrial | Bituminous Coal | " | " | 250 | 25.4 |
| Industrial | Subbituminous Coal | " | " | 343 | 25.9 |
| Industrial | Bituminous/Subbituminous Coal | " | " | 296 | 25.9 |
| Industrial | Coal | " | " | 289 | 25.9 |
| Industrial | Natural Gas | " | " | 81 | 14.5 |
| Industrial | Anthracite | " | " | 24 | 28.2 |
| Industrial | Waste Oil | 138.69¶ | e3gal | 14 | 20.0 |
| Industrial | Distillate Oil | " | " | 36 | 19.8 |
| Industrial | Residual Oil | " | " | 33 | 21.3 |
| Industrial | Liquified Petroleum Gas (LPG) [nonpoint/point] | " | " | 85◊/36 | 16.9 |
| Industrial | Coke [nonpoint/point] | " | " | 21/24 | 27.6 |
| Industrial | Process Gas | " | " | 33/10 | 15.3 |
| Industrial | Kerosene | 134.91§ | e3gal | 37 | 19.5 |
| Industrial | Jet Fuel | 120.19§ | e3gal | 54 | 19.2 |
| Industrial | Gasoline | 129.88§ | e3gal | 60820 | 19.2 |
| Industrial | distillate oil (no 2) | 139.93§ | e3gal | 36 | 19.8 |
| Industrial | Distillate Oil (Diesel) | " | " | 48 | 20.0 |
| Industrial | Refinery Gas | " | " | 33 | 15.3 |
| Industrial | jet A fuel | 120.19§ | e3gal | 54 | 19.2 |
| Industrial | jet naptha | 120.19§ | e3gal | 54 | 19.7 |
| Industrial | Oil | " | " | 36 | 19.8 |
| Industrial | blast furnace gas | 92⌐ | e6ft3 | 5554 | 56.3 |
| Industrial | coke oven gas | 574⌐ | e6ft3 | 1836 | 11.1 |
| Industrial | Propane | 90.42§ | e3gal | 35◊ | 17.0 |
| Commercial | Anthracite Coal | " | " | 240 | " |
| Commercial | Bituminous Coal | " | " | 250 | " |
| Commercial | Subbituminous Coal | " | " | 343 | " |
| Commercial | Bituminous/Subbituminous Coal | " | " | 296 | " |
| Commercial | Coal | " | " | 530 | " |
| Commercial | Natural Gas | " | " | 81 | " |
| Commercial | Distillate Oil | " | " | 36 | " |
| Commercial | Residual Oil | " | " | 33 | " |
| Commercial | Liquified Petroleum Gas (LPG) [nonpoint/point] | " | " | 85◊/21 | " |
| Commercial | Kerosene | " | " | 37 | " |
| Commercial | Diesel [nonpoint/point] | 137.06§ | e3gal | 36◊/929 | 20.0 |
| Commercial | Gasoline | " | " | 60820 | " |
| Commercial | Jet fuel | " | " | 19 | " |
| Commercial | Process gas | " | " | 33 | " |



| | | | | | |
|---|---|---|---|---|---|
| Commercial | Propane | " | " | 21$^\diamond$ | " |
| Commercial | Anthracite culm | 27.49$^\varsigma$ | tonne | 12 | " |
| Residential | Bituminous Coal | " | " | 11441 | " |
| Residential | Subbituminous Coal | " | " | 15705 | " |
| Residential | Bituminous/Subbituminous Coal | " | " | 13573 | " |
| Residential | Coal | " | " | 13238 | " |
| Residential | Anthracite | " | " | 11028 | " |
| Residential | Natural Gas [nonpoint/point] | " | " | 39$^\diamond$/63 | " |
| Residential | Distillate Oil | " | " | 36 | " |
| Residential | Residual Oil | " | " | 33 | " |
| Residential | Liquified Petroleum Gas (LPG) | " | " | 21 | " |
| Residential | Kerosene | " | " | 37 | " |
| Residential | Propane | " | " | 21 | " |
| Railroad | Diesel | " | " | 428$^\diamond$ | " |
| Railroad | Distillate Oil (diesel) | * | * | 811 | " |
| Marine vessels | Diesel | " | " | 428$^\diamond$ | " |
| Marine vessels | Residual Oil | " | " | 33 | " |
| Marine vessels | Gasoline | " | * | 60820 | " |
| Nonroad | Gasoline | " | * | 60820 | " |
| Nonroad | Distillate Oil (diesel) | " | * | 929 | " |
| Nonroad | Liquified Petroleum Gas (LPG) | " | * | 28 | " |

$^\varsigma$ Average heating value estimated at the state scale using a volume-weighted average. Source: Department of Energy/Energy Information Administration, Electric Power Monthly, 2011. Form EIA-423, "Monthly Cost and Quality of Fuels for Electric Plants Report;" Federal Energy Regulatory Commission, FERC Form 423, "Monthly Report of Cost and Quality of Fuels for Electric Plants."

$^\P$ Natural gas heat value is sourced to EPA 2018, Annex 2, Table A-46, page A-76. Petroleum fuels heat values sourced to EPA 2018, Annex 2, Table A-50, page A-83.

$^\S$ Values from Table 3-5, "Compendium of Greenhouse Gas Emissions Methodologies for the Oil and Gas Industry, American Petroleum Institute, February 2005. There is an updated document and it is API 2009: Table 3-8 on page 3-21.

$^f$ All values reported in Gurney et al., (2009) unless specified otherwise.

$^\diamond$ Value retrieved from self-reported data (see main text).

$^\Pi$ http://www.engineeringtoolbox.com/heating-values-fuel-gases-d_823.html

Fossil fuel $CO_2$ emissions are created from NEI-reported county-scale CO reporting through the application of CO and $CO_2$ emission factors as follows:

$$E_{n,f}^{CO_2} = \frac{E_{n,f}^{CO}}{EF_{n,f}^{CO}} EF_{n,f}^{CO_2} \qquad (1)$$

where $E_{n,f}^{CO_2}$, are the $CO_2$ emissions for a process $n$ (e.g. industrial 10 MMBTU boiler, industrial gasoline reciprocating turbine) and fuel $f$ (e.g. natural gas, bituminous coal); $E_{n,f}^{CO}$ are the equivalent amount of CO emissions for a process $n$ and fuel $f$; $EF_{n,f}^{CO}$ is the CO emission factor for a process $n$ and fuel $f$; and $EF_{n,f}^{CO_2}$ is the $CO_2$ EF for a process $n$ and fuel $f$. The CO EF is retrieved from two categories of source information: 1) "self-reported" values (supplied by state or federal air quality specialists submitting the CO emissions reporting: ftp://newftp.epa.gov/air/nei/2011/doc/2011v2_supportingdata/nonpoint/)[1] or 2) "default" values generated from a combination of values retrieved from the EPA WebFIRE EF database (https://cfpub.epa.gov/webfire/) and values accumulated through literature review (see Table 2 and table footnotes for details). The self-reported CO EF values are assessed for reliability and replaced by a default value if the self-reported value is less than 0.1 or greater than 5 times the identified default value.

---

[1] The file "NonPoint_Activity2011V2.csv" is no longer archived and/or available from the United States Environmental Protection Agency.



The state total $FFCO_2$ emissions calculated as described above are compared to sector and fuel-specific fuel
consumption totals reported by the Department of Energy/Energy Information Administration (DOE/EIA) State
Energy Data System (DOE/EIA, 2018). The EIA SEDS consumption data are gathered to create a historical time
series of energy production, consumption, prices and expenditures for members of congress, federal and state
agencies and the general public in addition to supporting EIA energy modeling analysis. The consumption in energy
units are converted to $FFCO_2$ using $CO_2$ EFs for each fuel type category (natural gas, petroleum, coal) from values
supplied in Table 2. Because the EIA SEDS does not separately report nonpoint versus point sources for a given
sector/fuel combination, the sum of the Vulcan nonpoint and point (see next section) $FFCO_2$ emissions are compared
to the EIA/SEDS totals. Adjustment of the Vulcan state/sector/fuel totals are made to the nonpoint residential and
commercial sectors only and for natural gas and petroleum fuel (aggregate) only. This is driven by the understanding
that the survey sampling performed by the EIA SEDS in the industrial sector is more uncertain due to the variety of
fuel consumption circumstances and idiosyncratic contractual arrangements made between utilities/fuel suppliers
and industrial entities. Furthermore, industrial facilities have the capability to "stockpile" fuel, making use of annual
consumption data difficult to interpret without stockpile information. This, and the fact that the coal-based emissions
are small to non-existent in the residential and commercial sectors, is why adjustment is not made for coal fuel
values. The adjustments made to the nonpoint residential and commercial $FFCO_2$ emission amounts are shown in the
supplementary material, Table S1.
Sub-county distribution of the county/sector/fuel-specific $FFCO_2$ emissions to US Census block-groups uses the
total floor area ($m^2$) of buildings (specific to a building class) within each US Census block-group combined with
estimates of energy use intensity (EUI). The general approach follows:
$$TE_{n^3,f}^{bg} = TFA_{n^1}^{bg} \times EUI_{n^2,f}^{cd} \{n^1 \rightarrow n^2 \rightarrow n^3\} \qquad (2)$$
where the total emissions, *TE*, associated with a building of type, *n*, using fuel, *f*, in a block-group, *bg*, is equal to the
product of the total floor area, *TFA*, and the energy use intensity, *EUI*, of buildings in a census division, *cd*. Because
the data sources have somewhat different building type classification schemes, a crosswalk between the various
categories must be achieved.
Building floor area is retrieved from HAZUS General Building Stock data collected and compiled by the Federal
Emergency Management Agency (FEMA, 2017). Using multiple sources including the US Census and the DOE, the
FEMA floor area provides an estimate of the building floor area for each US Census block-group specific to a
classification of building types in the residential, commercial and industrial sectors. The data sources are primarily
reflective of conditions in 2010.
The non-electric energy use intensity (NE-EUI; joules/$m^2$) values are compiled by the DOE from building
consumption energy surveys in different regions of the US. The NE-EUI values were calculated from data in the
DOE/EIA Commercial Buildings Energy Consumption Survey (CBECS, 2016), Manufacturing Energy
Consumption Survey (MECS, 2010), and Residential Energy Consumption Survey (RECS, 2013) microdata which
represent regional (9 US Census Divisions) surveys of building energy consumption categorized by building type,

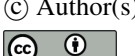



fuel, and age cohort. The three data sources represent survey conditions in 2012, 2009, and 2010, respectively. A
crosswalk is created linking the FEMA building types to the DOE/EIA building types (supplementary material,
Table S2). For the industrial sector, data is insufficient to support specificity to US Census Division. Hence, the
national average results are used but specific to industrial NAICS category and fuel category.
Where insufficient data existed to support Census Division-specific NE-EUI values in any of the three sectors, an
average was calculated using all other Division/building type/fuel-specific NE-EUI values.
The product of the total building area for a given Census block-group/sector/building type combination and the
sector/building type/fuel NE-EUI values act as a distributional fraction of the county total county/sector/fuel $FFCO_2$
to each Census block-group. Hence this acts to provide a relative distribution of building $FFCO_2$ emission within a
US county only.
The time distribution of the annual $FFCO_2$ emissions for the nonpoint data source uses a building energy model,
eQuest, to generate simulated building energy consumption which, in turn, is used to represent hourly time patterns
(Hirsch & Associates, 2004). The eQuest simulations are based on a series of building prototypes which must be
related to the FEMA building typology (in turn, related to the final Vulcan building types – see Table S2) of the
Vulcan system. This relationship is shown in supplementary material, Table S3.
To capture the local weather/climate conditions, the eQuest model is additionally driven by the 1020 "TMY3
weather station datasets (http://doe2.com/Download/Weather/TMY3/) from the DOE (Marion and Urban, 1995).
The weather statistics reflect the 1991-2005 climatological mean conditions. The resulting simulations are used to
generate hourly fractional energy consumption for each of the weather station locations and for each of the building
types listed in Table S3. The closest weather station location to each of the Census block-group centroids is used to
assign these hourly fractional time series to a given block-group/building type combination.
*Uncertainty*
Nonpoint source uncertainty is applied to the reported CO emissions, the CO EF, and the $CO_2$ EF. For the reported
CO emissions, an uncertainty value of ±12.8% was used, a value reported by Gately et al. (2017) for the residential
sector (which dominates the nonpoint sources) and based on a state-scale difference between ACES and EIA state
residential fuel consumption. We interpret this as a 95% confidence interval given that this is estimated from a
measure of difference by Gately et al. (2017).
For the EF uncertainty, the CO and $CO_2$ EFs were adjusted in combination such that the outcome achieves the hi and
lo CI, respectively. For example, the upper/lower CI bound for the CO EF was combined with the lower/upper CI
bound for the $CO_2$ EF to achieve the hi/lo $FFCO_2$ emissions output CI bound.
An uncertainty of ±20% is applied to both the default and self-reported CO EFs regardless of fuel type. An
exception to this is for the "blast furnace gas" and "coke oven gas" fuel types in which the adjustment is ±35%
(Table 3). The CO EF adjustment is based on estimates of the range found in the WebFIRE database and the self-
reported CO emission factors. The $CO_2$ EF uncertainty for coal is derived from the work of Quick (2010) while





1   uncertainty for petroleum fuels and natural gas are derived from USEPA Greenhouse Gas Inventory, Annex 2

2   (USEPA, 2017).

3   **Table 3: Upper and lower confidence interval values for the CO and CO₂ emission factors.**

| Sector | Fuel | CO EF lo/hi (lbs/e9btu)/ | CO₂ EF lo/hi (tC/e9btu) |
|---|---|---|---|
| Electricity Production | Bituminous Coal | 296 / 196 | 24.7 / 26.1 |
| Electricity Production | Subbituminous Coal | 413 / 275 | 25.5 / 26.4 |
| Electricity Production | Bituminous/Subbituminous Coal | 354 / 236 | 25.1 / 26.8 |
| Electricity Production | Coal | 34.8 / 23.2 | 23.8 / 28.0 |
| Electricity Production | Anthracite | 28.8 / 19.2 | 26.3 / 30.0 |
| Electricity Production | Lignite | 46.8 / 31.2 | 25.4 / 27.1 |
| Electricity Production | Natural Gas | 75.6 / 50.4 | 13.8 / 15.2 |
| Electricity Production | Distillate Oil | 43.2 / 28.8 | 19.0 / 20.5 |
| Electricity Production | Residual Oil | 39.6 / 26.4 | 19.2 / 23.4 |
| Electricity Production | Liquified Petroleum Gas (LPG) | 33.6 / 22.4 | 15.9 / 17.9 |
| Electricity Production | Process Gas | 39.6 / 26.4 | 11.3 / 19.3 |
| Electricity Production | Coke | 25.2 / 16.8 | 25.9 / 29.2 |
| Electricity Production | Distillate Oil (Diesel)/Diesel | 1114 / 743 | 19.3 / 20.8 |
| Electricity Production | Oil | 43.2 / 28.8 | 17.8 / 21.8 |
| Electricity Production | Jet Fuel | 901 / 601 | 18.4 / 19.9 |
| Electricity Production | Refinery Gas | 39.6 / 26.4 | 11.3 / 19.3 |
| Industrial | Bituminous Coal | 300 / 200 | * |
| Industrial | Subbituminous Coal | 412 / 274 | * |
| Industrial | Bituminous/Subbituminous Coal | 355 / 237 | * |
| Industrial | Coal | 347 / 231 | * |
| Industrial | Natural Gas | 97.2 / 64.8 | * |
| Industrial | Anthracite | 28.8 / 19.2 | * |
| Industrial | Waste Oil | 16.8 / 11.2 | 18.0 / 22.1 |
| Industrial | Distillate Oil | 43.2 / 28.8 | * |
| Industrial | Residual Oil | | * |
| Industrial | Liquified Petroleum Gas (LPG) [nonpoint/point] | 101 / 67.8° 43.4 / 28.9 | * |
| Industrial | Coke [nonpoint/point] | 25.2 / 16.8 43.4 / 28.9 | * |
| Industrial | Process Gas | 39.6 / 26.4 120 / /80.3 | * |
| Industrial | Kerosene | 44.4 / 29.6 | 18.8 / 20.3 |
| Industrial | Jet Fuel | 648 / 432 | * |
| Industrial | Gasoline | 72984 / 48656 | 18.3 / 20.0 |
| Industrial | distillate oil (no 2) | * | 19.0 / 20.5 |
| Industrial | Distillate Oil (Diesel) | 57.6 / 38.4 | * |
| Industrial | Refinery Gas | 39.6 / 26.4 | * |
| Industrial | jet A fuel | 648 / 423 | 18.4 / 19.9 |
| Industrial | jet naptha | 648 / 432 | 18.9 / 20.5 |
| Industrial | Oil | * | * |
| Industrial | blast furnace gas | 7498 / 3610 | 41.5 / 71.0 |
| Industrial | coke oven gas | 2479 / 1194 | 8.18 / 13.9 |
| Industrial | propane | 42.5 / 28.3° | 16.0 / 18.1 |
| Commercial | Anthracite Coal | * | * |
| Commercial | Bituminous Coal | 300 / 200 | " |
| Commercial | Subbituminous Coal | 412 / 274 | " |
| Commercial | Bituminous/Subbituminous Coal | 355 / 237 | " |
| Commercial | Coal | 636 / 424 | " |
| Commercial | Natural Gas | 97.2 / 64.8 | " |
| Commercial | Distillate Oil | * | " |
| Commercial | Residual Oil | * | " |
| Commercial | Liquified Petroleum Gas (LPG) [nonpoint/point] | 102 / 67.8° 25.5 / 17.0 | " |
| Commercial | Kerosene | * | " |
| Commercial | Diesel [nonpoint/point] | 43.8 / 29.2° 111 / 74.3 | * |
| Commercial | Gasoline | * | " |



| Commercial | Jet fuel | * | " |
|---|---|---|---|
| Commercial | Process gas | * | " |
| Commercial | Propane | 25.2 / 16.8$^{\diamond}$ | " |
| Commercial | Anthracite culm | 14.4 / 9.62 | " |
| Residential | Bituminous Coal | 13729 / 9153 | " |
| Residential | Subbituminous Coal | 18846 / 12564 | " |
| Residential | Bituminous/Subbituminous Coal | 16288 / 10858 | " |
| Residential | Coal | 15886 / 10590 | " |
| Residential | Anthracite | 13234 / 8822 | " |
| Residential | Natural Gas [nonpoint/point] | 46.5 / 31.0$^{\diamond}$ 79.6 / 50.4 | " |
| Residential | Distillate Oil | * | " |
| Residential | Residual Oil | * | " |
| Residential | Liquified Petroleum Gas (LPG) | 25.5 / 17.0 | " |
| Residential | Kerosene | * | " |
| Residential | Propane | * | " |
| Railroad | Diesel | 514 / 343$^{\diamond}$ | " |
| Railroad | Distillate Oil (diesel) | 973 / 649 | " |
| Marine vessels | Diesel | 514 / 343$^{\diamond}$ | " |
| Marine vessels | Residual Oil | 39.6 / 26.4 | " |
| Marine vessels | Gasoline | 72984 / 48656 | " |
| Nonroad | Gasoline | 72984 / 48656 | " |
| Nonroad | Distillate Oil (diesel) | 1115 / 743 | " |
| Nonroad | Liquified Petroleum Gas (LPG) | 33.6 / 22.4 | " |

### 2.1.3 Point data

The point emissions represent facilities with a physically identifiable emission "stack" or point location and exceed a specific criteria air pollution threshold (USEPA 2015c). The NEI point source data files are primarily comprised of processes associated with the industrial and airport sectors but emissions from the commercial, railroad, nonroad, and electricity production sectors are present as well (USEPA 2015a).

A number of key fields that define a point location for the purposes of the Vulcan $FFCO_2$ emissions estimation within the point database and include the state and county FIPS code, the "state facility identifier" (which identifies the individual emitting facility) and the tribal code (used in place of the FIPS in tribal lands). Each site or facility can have multiple emission points (different "stacks"), units (different buildings or portions of a complex facility or site), or emission processes (e.g. energy production, heaters, engines). Some of the emitting points/units/processes can have different geocoded locations and these are retained in the Vulcan processing. Hence, exact latitude and longitude is critical for allocation to the physical US landscape. Corrections to location information were made in urban domains associated with the Hestia Project : the Los Angeles Basin, Baltimore, Salt Lake City, and Indianapolis (e.g. Gurney et al., 2018; 2019b).

Each point emission record is also associated with an SCC which is used to retrieve the needed CO and $CO_2$ EFs to enact the same procedure outlined in the description of the nonpoint source processing. In the case of the point sources, no self-reported EFs are supplied. Separation is first made between airport point sources (processing of which is described in a later section) and non-airport point sources. The non-airport point sources are matched to a CO EF via the SCC from the EPA's WebFIRE EF database as the first choice for the CO EF. Where no match is found, default CO EF values are used, themselves archived from literature review (see Table 2) and determined through a combination of the sector and fuel.



All point source emission records designated as industrial, railroad, and nonroad are distributed to hourly temporal
resolution from the 2011 annual total using SCC-specific temporal surrogate profiles provided by the EPAs
Clearinghouse for Inventories and Emissions Factors (CHIEF) (USEPA, 2015c). The temporal surrogate profiles are
constructed from monthly, weekly and diurnal cycles (data available at:
ftp://newftp.epa.gov/air/emismod/2011/v3platform/ancillary_data/ge_dat_for_2011v3_temporal.zip). These
temporal surrogates are comprised of three cyclic time profiles (diurnal, weekly, monthly) specific to SCC that are
combined to generate hourly SCC-specific time fractions for an entire calendar year. Records which do not have an
SCC match are distributed as a constant hourly emission.
*Uncertainty*
Point source uncertainty is applied to the reported CO emissions, the CO EF, and the $CO_2$ EF. For the reported CO
emissions, an uncertainty value of ±7.8% is used, a value reported by Gately et al. (2017) for the industrial and
commercial sectors (which dominate the point sources) and based on a state-scale difference between ACES and
EIA state industrial+commercial fuel consumption. We interpret this as a 95% confidence interval given that this is
estimated from a measure of difference by Gately et al. (2017).
For the default EF uncertainty, the CO and $CO_2$ EFs were adjusted in combination in a fashion similar to that
described in the nonpoint source section and the same percentage numerical boundaries described there were used.
For the records that use the WebFIRE CO EFs, an uncertainty value of ±20% is used for the 95% CI bounds.
**2.1.4    Electricity Production**
Three sources of data are used to estimate the $FFCO_2$ emissions at electricity production facilities, all are geocoded
to a physical location. The first is the Environmental Protection Agency's Clean Air Markets Division (CAMD) data
(USEPA, 2015b). The second is the Department of Energy's Energy Information Administration (EIA) reporting
data (DOE/EIA, 2003). The third is the reporting done within the NEI point source reporting (described previously).
Overlap exists between these three data sources (corrected in the processing here) which is corrected according to
the prioritization in the order listed above. A detailed comparison made between the CAMD and EIA $FFCO_2$
emissions along with greater detail regarding data sources, data processing and procedures can be found in Gurney
et al. (2016).
The CAMD data is collected under the Acid Rain Program (ARP), which was instituted in 1990 under Title IV of
the Clean Air Act (CFR 2008; USEPA 2005b; USEPA 2010). Though the CAMD dataset does not include all power
plants in the US, it accounts for a very large proportion. The CAMD data used in Vulcan are reported as hourly $CO_2$
emissions monitored from an emitting stack or through a calculation, based on records of fuel consumption
(ftp://ftp.epa.gov/dmdnload/emissions/hourly/monthly/). The annual reporting is also used for additional information
related to the facility (http://ampd.epa.gov/ampd).
The EIA dataset is derived from the EIA reporting form 923, which reports monthly data on receipts and cost of
fossil fuel, fuel stocks, generation, consumption of fuel for generation, and environmental data at each power plant



(http://www.eia.gov/electricity/data/eia923). Fuel consumption is reported as a heat input value (e.g. british thermal
units). $CO_2$ emission factors are then utilized to calculate the quantity of $CO_2$ emitted. In order to maintain
consistency with the data source, the $CO_2$ emission factors used by the EIA are adopted to estimate the $FFCO_2$
emissions from these facilities (DOE, 2011).
Some manual corrections are performed to the geocoordinates of both the CAMD and EIA electricity production
data, as a result of searching in Google Earth or via alternative online information resources (e.g. utility websites).
A hierarchy was employed given that there was overlap between the two datasets. This was performed at the unit
level given that a single facility might have individual power units reporting to CAMD and another only reporting to
the EIA. Where overlap did exist at this scale, preference was made to retain the CAMD data. Further details and
rationale can be found in Gurney et al. (2016).
The CAMD reporting data is archived at the hourly temporal scale and directly used in Vulcan. The EIA electricity
production reporting is resolved at the monthly scale. This is transformed into hourly reporting using a "flat" time
profile or a constant level such that the monthly integral matches the reported monthly emissions data. The
electricity production facilities reported in the NEI as point sources also use a flat time profile but instead of
distribution over each of the reported months, the emissions are held constant over an entire year.
Table 4 provides a summary of the electricity production data totals for the three data sources.
**Table 4: Summary information for 2011 electricity production facilities in Vulcan version 3.0.**

| Data source | Number of facilities | Total FFCO2 emissions (MtC/year) |
|---|---|---|
| CAMD | 1479 | 592.1 |
| EIA | 2255 | 40.00 |
| NEI | 11832 | 8.87 |
| Total | 15566 | 641.0 |

*Uncertainty*
Gurney et al., (2016) found that one-fifth of US power plants had monthly $FFCO_2$ emission differences exceeding -
6.4%/+6.8% for the year 2009 (the closest analyzed year to the 2011 base year presented here). The emissions
distribution of the two datasets were not normally distributed nor were the differences. Hence, a typical gaussian
uncertainty estimate cannot be made – rather, the difference distribution was represented by quintiles of percentage
difference. Hence, these values cannot be cast within the context of other normally-distributed errors. However, we
conservatively consider the quintile value (the positive and negative tails) as a one-sigma value and ±13% as a 95%
CI boundary value.
**2.1.5    Onroad**
County scale $FFCO_2$ emissions are retrieved from the 2011 EPA NEIv1 onroad results (USEPA, 2011). The 2011
NEI onroad results report emissions for every US county by 13 vehicle types (designating vehicle class and fuel)
and 12 road types, including urban and rural distinctions. It is based on simulations using the Motor Vehicle



Emissions Simulator (MOVES) model with inputs supplied to a county database (CDB) by SLT agencies (USEPA,
2012; USEPA, 2015a). Version 1.0 of the 2011 NEI includes 1,363 CDB submissions out of a total of 3,234
counties. In order to generate results for all counties in the US, the EPA used multiple data and modeling tools to
estimate county-specific $FFCO_2$ emissions including identifying "representative" counties among the data supplied
by SLT agencies to best match those there were not reported.
The state of California did not report $FFCO_2$ to the 2011 NEI. Hence, the Vulcan onroad $FFCO_2$ emissions for
California used the 2011 results from the Emissions FACtors 2014 model (EMFAC2014), produced by the
California Air Resources Board (CARB, 2014). The EMFAC2014 model estimates vehicle miles traveled (VMT)
and $FFCO_2$ emissions for 27 vehicle types (reduced here to 13 via aggregation) using emissions rates
($FFCO_2$/distance traveled) and data on the California vehicle fleet and activity statistics such as VMT, speed
distributions, and idle times (CARB, 2015). Distribution to sub-state scales uses annual vehicle counts from the
Highway Performance Monitoring System (HPMS). The HPMS is a spatial road network database managed by the
FHWA to monitor and record Average Annual Daily Traffic (AADT) counts (FHWA, 2014). By considering
vehicle registration in combination with the HPMS data, EMFAC also accounts for inter-regional travel.
County-scale $FFCO_2$ emissions for all US states are spatially assigned to road segments via a road basemap that best
represents the entirety of the road surface occupied by onroad vehicles. Vulcan uses a combination of the 2011
Highway Performance Monitoring System (HPMS, 2017) road network and Open Street Map (OSM;
http://download.geofabrik.de/) road network. The Census Urbanized Areas boundary
(https://www.fhwa.dot.gov/policyinformation/hpms/shapefiles.cfm) was used to assign an urban/rural distinction to
each of the 7 original HPMS road classes making them compatible with the onroad NEI road classes (supplementary
material, Table S4).
The distribution of the county-scale road/vehicle-specific $FFCO_2$ emissions along the complete length of road class
in a county, is achieved through the use of the 2011 AADT data from the FHWA's HPMS
(http://www.fhwa.dot.gov/policyinformation/hpms/shapefiles.cfm; state scale data files were used). AADT counts
are collected using short-term and continuous counting methods. Most data are collected by individual states and
reported to the FHWA, but some data are also collected by the FHWA directly. Very little AADT data was collected
on local roads (urban local, rural local). For those segments in our merged basemap that do not have an AADT
value, gap-filling was used (see supplementary material for details on gap-filling methods).
With a complete US map of AADT values and road segment length, the vehicle miles traveled (VMT) can be
estimated. The fraction of a non-local road class-specific road segment's VMT within a county acts as the
distribution means to allocate county-scale onroad $FFCO_2$. For local roads, given the paucity of AADT data, the
fraction of a road segment's length out of all local roads within a county acts as the allocation method. Hence, the
local roads have no spatial gradients along the local roads (at the sub-county scale). However, there are $FFCO_2$
emissions gradients in space that are determined by the spatial density of local roads.
In order to use the spatial distribution methods employed by the Vulcan system and be compatible with the NEI
results for the other US states, the vehicle class/county-specific California onroad $FFCO_2$ emissions must be





translated to the 6 vehicle classes and 14 road classes (7 of urban and rural sub-types) in the NEI. This is performed
via the use of Federal Highway Administration (FHWA) state-scale VMT data by road class and the proportion of
VMT by vehicle class. Details are provided in the supplementary material.
Carrying out the spatialization procedure across all counties in the United States, it became clear that there were
some mismatches between NEI road class VMT and the AADT on the HPMS road network. For example, there
were instances in which onroad FFCO$_2$ emissions were present in a county for a particular road class, but for which
no AADT data existed and vice-versa. These mismatches could be due to the demarcation of urban versus rural
roads. As noted previously the roads were divided into urban and rural classes based on the US Census Urbanized
Areas. This may differ from the choices made when state officials were generating the county database inputs for the
EPA (if the NEI estimate uses state-supplied data in the MOVES onroad emissions estimate, for example). While
the HPMS AADT data has an urban code, we used the US Census Urbanized Areas to divide a road classes so that
the urban/rural classification would be consistent between the OSM and HPMS basemaps.
In cases where emissions were reported for a road class in NEI, but for which there were no physical roads in our
AADT gap-filled basemap, the emissions reported in NEI were moved to the next closest road class with AADT.
The closest road class is the urban or rural counterpart within the same class-size, and the second-closest being the
road class that is the next class-size down. In cases where AADT was present for a road class, but no NEI FFCO$_2$
emissions were reported for that road class, FFCO$_2$ emissions were redistributed from the next closest road class,
proportional to VMT. For example, if the NEI reports emissions for urban interstates, but VMT was estimated for
both urban and rural interstates, then the NEI reported emissions would be redistributed from urban interstates to
rural interstates proportional to the VMT in each road class.
In the state of California, the EMFAC results were crosswalked from county-scale, vehicle class-specific FFCO$_2$
emissions to totals that include road class. These FFCO$_2$ emissions were distributed onto road segments in the same
manner as done for other states. However, unlike other states, there were no cases in which the EMFAC onroad
FFCO$_2$ emissions needed to be "shuffled" to partner road classes.
Hourly traffic volume data for the years 2011-2013 were obtained from the FHWA Continuous Count Stations
(CCS) dataset (previously known as the Automatic Traffic Recorder; ATR) (Jessberger, 2016). The CCS stations
measure hourly traffic volume at a fixed location in space and we use the station's latitude and longitude as a unique
station identifier. Corrections were made to the Connecticut station coordinates based on data from the Connecticut
Department of Transportation (http://www.ct.gov/dot/cwp/view.asp?a=1383&q=330402).
For each station, the direction(s) and lane(s) of traffic are recorded but are aggregated to estimate the total traffic
volume moving through a station across all lanes and directions. The result is a single measure of traffic volume per
hour that is traveling through a station in any direction and on any lane with a unique location. Any station with
greater than six months total (either contiguous or not) of missing traffic monitoring data, were removed from the
dataset. This left a total of 5106 traffic volume monitoring stations in the year 2011, 5172 in 2012 and 5527 in 2013.
2011 contained 141 stations that were not present in either 2012 or 2013. 2012 contained 57 stations that were not
present in either 2011 or 2013. 2013 contained 511 stations that were not present in either 2011 or 2012. Each year



of the traffic monitoring data (for which there are no instances of gaps exceeding six months) are gap-filled
individually, maintaining the cyclic integrity of the hour of day and day of the week. Details are provided in the
supplementary material.
After combining the 2011, 2012 and 2013 CCS data into a single average year dataset, there were a total of 6047
CCS measurement locations including Alaska and Hawaii. There are 5890 stations in the Continental US and these
are used for the construction of the temporal profiles.
In order to distribute the temporal distribution measured at the gap-filled CCS measurement stations to all road
segments in the US, interpolation/extrapolation of the traffic patterns is required. Given the paucity of traffic
measurement stations relative to the total area of the US landscape and the fact that the temporal distribution of
traffic is less related to road class than space, it was determined to aggregate the eight road classes to four,
"temporal" road classes for purposes of spatial interpolation. There is evidence that interstates have unique traffic
patterns from all other road classes due to the preponderance of interstate trucking commerce. Furthermore,
interstate usage in cities is a mix of passenger vehicles and commercial trucking while rural interstates are
dominated by commercial trucking. Hence, the road classes chosen for the purposes of temporal interpolation were:
rural interstate, urban interstate, rural non-interstate, and urban non-interstate. Figure 1 shows the CCS measurement
locations aggregated to these four temporal road classes.

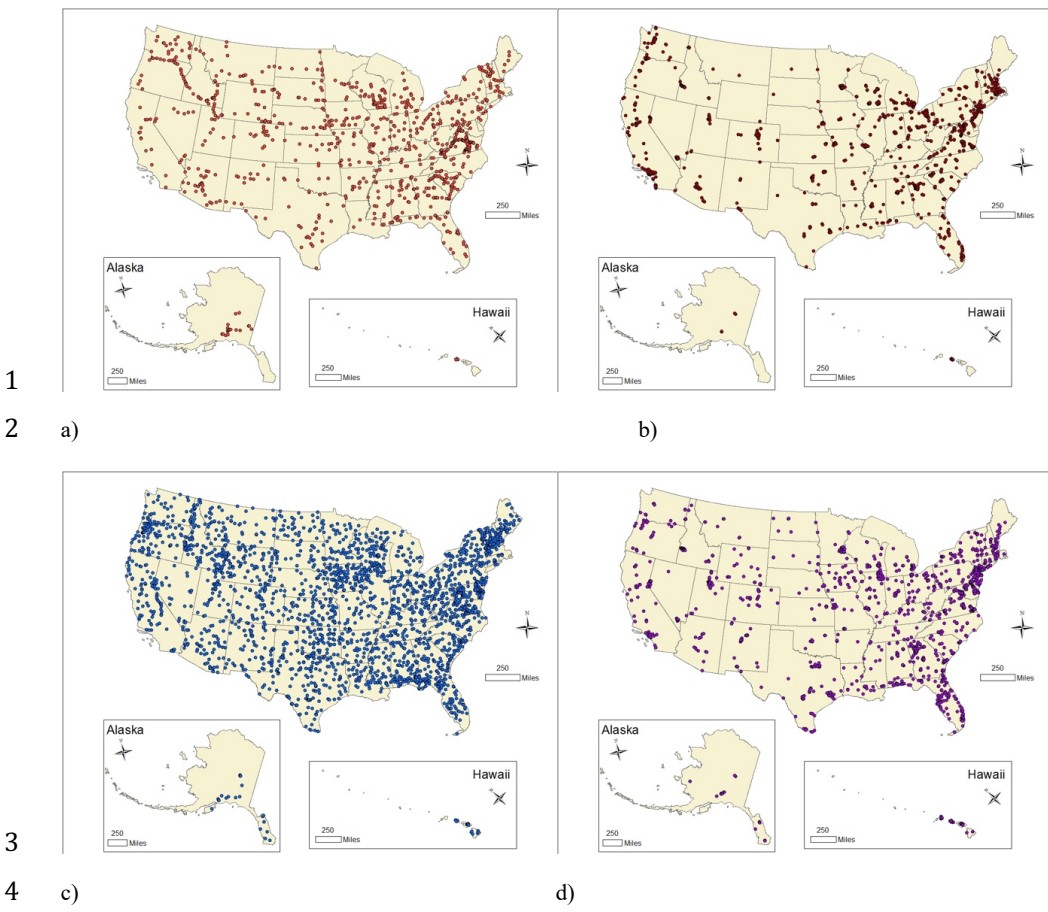

a)                                                                                        b)
c)                                                                                        d)

**Figure 1: Distribution of CCS measurements stations separated into four road classes. a) rural interstate; b) urban interstate; c) rural non-interstate; d) urban non-interstate.**

Inverse Distance Weighted (IDW) interpolation was performed for each of the four temporal road classes separately,
and only for grid cells that are occupied by roads of that road class. The two inputs are the gap-filled CCS traffic
data, and the locations of road segments for each of the four road classes. The IDW used the default number of
neighbors (all neighbors), and the default power function (2), making this an inverse distance squared method.
*Uncertainty*
The uncertainty in the onroad sector uses the results from Gately et al., (2017) which, in turn, references Gately et
al., (2013) and Mendoza et al., (2013). This uncertainty was estimated at ±7.1% for a presumed 1-sigma uncertainty.
Here, we have assigned ±14.2% to the 95% CI boundaries for all road types.





### 2.1.6    Nonroad

The nonroad sector $CO_2$ emissions estimates are retrieved from the 2011 EPA NEIv2 which uses the NONROAD model to estimate emissions (ftp://ftp.epa.gov/EmisInventory/2011/2011neiv2_nonroad_byregions.zip) across a large number of mobile sources that travel "off-road" (USEPA, 2015a) except locomotives, airplanes and commercial marine vessels (CMV) which are taken up in separate sections in this document. The NONROAD model results, in turn, are based on output from the National Mobile Inventory Model (NMIM) which relies on data inputs from the National County Data base (NCD) (USEPA 2005c; 2005d). Both the NMIM and the NCD were described previously (Gurney et al., 2009). The EPA updated data within the NCD from 12 SLT agencies along with EPA default values to generate the results in the 2011 NEIv2 (for a description of these updates see ftp://ftp.epa.give/EmisInventory/2011/doc/2011neiv2_supdata_nonroad).

As with the onroad sector, California presents a special case. The CO emissions are reported comprehensively using California's OFFROAD model (www.arb.ca.gov/msei/offroad/offroad.htm ) but no $CO_2$ was reported. Hence, we scaled the California CO emissions by the mean SCC-specific $CO_2/CO$ ratio from all other US counties.

Spatial distribution uses the spatial surrogates generated by the EPA reflecting a series of spatial representations such as the mines, golf course and agricultural land (The shapefiles can be found here: ftp://ftp.epa.gov/EmisInventory/emiss_shp2003/us/ or ftp://ftp.epa.gov/EmisInventory/2011v6/v1platform/spatial_surrogates/shapefiles/). There were instances in which nonroad $FFCO_2$ emissions could not be associated with a spatial entity due to missing data. These emissions are spatialized by first aggregating all the unassociated sub-county emission elements to the county scale for a given spatial shape (e.g., golf courses, mines) and then distributing these emissions evenly across the county.

The sub-annual temporal distribution of the nonroad $FFCO_2$ emissions uses SCC-specific temporal surrogate profiles provided by the EPAs Clearinghouse for Inventories and Emissions Factors (CHIEF) (USEPA, 2015c). The temporal surrogate profiles are constructed from monthly, weekly and diurnal cycles (data available at: ftp://newftp.epa.gov/air/emismod/2011/v3platform/ancillary_data/ge_dat_for_2011v3_temporal.zip).

These temporal surrogates are comprised of three cyclic time profiles (diurnal, weekly, monthly) specific to SCC that are combined to generate hourly SCC-specific time fractions for an entire calendar year. There are 5 SCC codes present in the NEI 2011 nonroad data file but not found in the temporal surrogate files (2260006035, 2265006035, 2267006035, 2270006035, 2268006035) - these were given a "flat" or constant time profile in the absence of any specified temporal distribution.

*Uncertainty*

Nonroad records other than those derived from the point source data files (which follow the point source uncertainty estimation described in the point source section) are assigned a 95% CI boundary of ±3.8% for the $FFCO_2$ emission value. This was derived from examination of the range of carbon content and fuel density uncertainties as outlined in EPA (2017), Annex 2, page A-86. This is consistent with the point source uncertainty for nonroad distillate fuel consumption.





### 2.1.7 Airport
As described in the point source section, the airport $FFCO_2$ emissions are estimated from the 2011 NEI point source
reporting for CO. The emission factors used (Table 5) convert the reported CO emissions to $FFCO_2$ and are specific
to aircraft class and fuel, consistent with the reporting in the NEI which often listed multiple processes (aircraft
class/fuel) for a single airport facility. The fuel type implied by the $CO_2$ EF values uses jet fuel except where
explicitly indicated in the SCC description (NG, LPG, diesel, gasoline).
**Table 5: SCC, description, CO EF and CO₂ EF values for the airport sources.**

| SCC | description | CO EF[Υ] (lbs/e6btu) | CO₂ EF (tC/e6btu) |
|---|---|---|---|
| 2275060011 | Aircraft /Air Taxi /Piston | 0.751 | 0.019 |
| 2275060012 | Aircraft /Air Taxi /Turbine | 0.751 | 0.019 |
| 2275070000 | Aircraft /Aircraft Auxiliary Power Units /Total | 0.5396 | 0.019 |
| 2268008005 | Airport Ground Support Equipment, CNG | 0.081 | 0.014 |
| 2267008005 | Airport Ground Support Equipment, LPG | 0.085 | 0.017 |
| 2270008005 | Airport Ground Support Equipment, Diesel | 0.828 | 0.020 |
| 2265008005 | Airport Ground Support Equipment, 4-Stroke Gasoline | 30.34 | 0.019 |
| 2275020000 | Aircraft /Commercial Aircraft /Total: All Types | 1.083 | 0.019 |
| 2275050011 | Aircraft /General Aviation /Piston | 0.751 | 0.019 |
| 2275050012 | Aircraft /General Aviation /Turbine | 0.751 | 0.019 |
| 2275001000 | Aircraft /Military Aircraft /Total | 1.083 | 0.019 |
| 27505011 | Aircraft /Civil /Jet Engine: Jet A | 1.083 | 0.019 |
| 27505001 | Aircraft /Civil /Piston Engine: Aviation Gas | 0.751 | 0.019 |
| 27502011 | Aircraft /Commercial /Jet Engine: Jet A | 1.083 | 0.019 |
| 27501015 | Aircraft /Military /Jet Engine: JP-5 | 1.083 | 0.019 |
| 2275060011 | Aircraft /Air Taxi /Piston | 0.751 | 0.019 |

[Υ] CO emission factors are retrieved from the Intergovernmental Panel on Climate Change Guidelines on National Greenhouse Gas Inventories (IPCC 2006). The values reflect a 50/50 mixture of new versus old aircraft fleet characteristics in addition to a 67/33 mixture of domestic and interntional flight characteristics. The emission factors for domestic aviation have been derived from an average of a number of typical aircraft. For domestic aircraft, the average fleet is represented by Airbus A320, Boeing 727, Boeing 737-400, Mc Donell Douglas DC9 and MD 80 aircraft. The old fleet is represented by Boeing B737 and Mc Donell Douglas DC9. For international traffic the average fleet is represented by Airbus A300, Boeing 767, B747 and Mc Donell Douglas DC10, whilst the old fleet is represented by the Boeing B707, Boeing 747 and Mc Donell Douglas DC8.

The airport $FFCO_2$ emissions are only associated with the taxi & takeoff/landing sequences. $FFCO_2$ emissions
associated with non-aircraft processes such as building operations and non-aircraft mobile sources are reported as
emissions in other sectors (e.g. commercial, nonroad). The airports are geocoded to the airport location in the NEI
though some manual adjustments have been made to the original coordinates using manual inspection in Google
Earth. The emission point, in these instances, is placed in the middle of the central runway.
Temporal distribution of the $FFCO_2$ airport emissions use a series of datasets. The Los Angeles World Airports
(LAWA) dataset reports hourly flight volume for three airports in the LA Basin domain: Los Angeles International
airport (LAX), Ontario airport (ONT), and Van Nuys airport (VNY) (Hastings, 2014). The Operations Network
(OPSNET) dataset from the FAA reports total date-specific, daily flight volume (365 values) at specific airports
(https://aspm.faa.gov/opsnet/sys/Default.asp). An hourly time profile was constructed by combining the LAWA
diurnal profile and the OPSNET annual profile. The three LAWA airports constituted the diurnal cycle (Figure 2) at
all US airports with the LAX assigned to international airports, the ONT to non-international airports and the VNY
to local airports.
Airports were matched with a Federal Aviation Administration (FAA) international airport database (FAAINTL) by
airport code to determine whether an airport is international



(https://hub.arcgis.com/datasets/4782d6f5aa844591a16d46df635b7af4_1). Airports which could not be matched to
the OPSNET data by airport code/airport name were assigned a temporal invariant ("flat") hourly time structure.

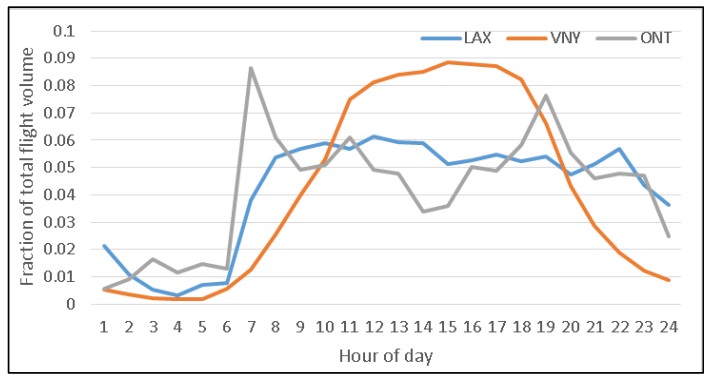

**Figure 2: Average hourly flight volume fractions at LAX, VNY and ONT**
FAAINTL, OPSNET, and two additional airport databases (the National Airport Atlas (NAA;
https://catalog.data.gov/dataset/airports-of-the-united-states-direct-download) and AIRNAV; www.airnav.com)
were used to determine whether an airport was an airport or a helipad. The name/code of each airport was searched
in these airport databases. An airport which could not be identified in any of the aforementioned airport databases
would be categorized as a helipad. A temporally invariant time structure was applied to all helipads.
A portion of the Vulcan v3.0 CMV $FFCO_2$ emissions would be considered "bunker" fuel combustion (i.e. consumed
as part of international travel) under the IPCC reporting methodology within the UNFCCC process. Vulcan does not
separate bunker from non-bunker fuel consumption and a portion of the airport sector emissions (particularly
international air flights) would be considered as such were the IPCC reporting categorization applied here. No
attempt has been made to limit or seperately report airport emissions that would be considered part of the bunker
fuel definition.
*Uncertainty*
The uncertainty in the airport sector is derived from the point source processing as described previously (magnitude
and EF-based uncertainty) except that the $CO_2$ EFs are specific to the mix of aviation fuels associated with the
emission records and are based on uncertainty estimation from the USEPA (2017), Annex 2, pages 85 & 89.
**2.1.8    Railroad**
The $FFCO_2$ emissions associated with railway activity are derived from the 2011 NEIv2 CO emissions reporting
which, in turn, were developed for the 2008 NEI (ERG 2011) and scaled to 2011 values (ERG 2012). Emissions
related to the railroad sector were reported as a mixture of nonpoint and point emissions and hence, these were
managed separately but combined when represented as spatial entities. The CO emissions were converted to $FFCO_2$
following the procedures outlined in the nonpoint and the point sections, respectively.



The two NEI source categories imply different spatial representations, however. The point source railroad emissions are associated with rail yards and related geo-specific locales and are placed in space according to the provided latitude and longitude. The railroad $FFCO_2$ emissions associated with the nonpoint NEI reporting contain an ID variable that links to a spatial element (rail line segment) in the EPA railroad GIS shapefile (https://www.epa.gov/sites/production/files/2015-06/railway_20140730.zip). A large number of railroad emission records have no railroad segment match and are spatialized using freight statistics described in supplementary material.

The annual railroad $FFCO_2$ emissions are distributed to the hourly timescale with no additional temporal structure (a "flat" time distribution), unless they originated from point source data for which the SCC-specific time profiles, previously described, are used.

*Uncertainty*

The uncertainty for the railroad emissions is directly inherited from the uncertainty estimation described for the point and nonpoint source processing, respectively. The only difference related to the the CO magnitude uncertainty ($\pm$3.8%) which was derived from examination of the range of carbon content and fuel density uncertainties outlined in EPA (2017), Annex 2, page A-86 for distillate fuels, the dominant fuel used in railroad.

### 2.1.9 Commercial Marine Vessels

The $FFCO_2$ emissions associated with commercial marine vessels (CMV) rely on nonpoint NEIv2 CO emissions reporting and follow the same emission factor-related conversion outlined in the nonpoint source section. CMV includes vessels directly or indirectly involved in commerce or military activity. The emissions encompass maneuvering, hoteling, cruise and reduced speed zone travel and are specific to geographically located ports and shipping lanes that extend 12 nautical miles from the US shoreline. Private or "pleasure" craft are not included as part of the CMV emissions but are captured in the nonroad reporting. As with the nonroad reporting, the EPA used a mixture of SLT data submissions and default values, in collaboration with the Office of Transportation and Air Quality to generate an estimate of CO emissions for CMV. A portion of the Vulcan v3.0 CMV $FFCO_2$ emissions would be considered "bunker" fuel combustion (i.e. consumed as part of international trade) under the IPCC reporting methodology within the UNFCCC process. Vulcan does not separate bunker from non-bunker fuel consumption and a portion of the CMV sector emissions (particularly ship travel directed towards international waters) would be considered as such were the IPCC reporting categorization applied here. No attempt has been made to limit or seperately report CMV emissions that would be considered part of the bunker fuel definition.

The spatialization utilized the EPA shapefiles that delineate US ports and US shipping lanes through spatial IDs associated with the emission records (https://www.epa.gov/sites/production/files/2015-06/ports_20140729.zip; https://www.epa.gov/sites/production/files/2015-06/shippinglanes_072914.zip). In the instance that no spatial entity is identified for an emission record, a simple spatial alternative is employed whereby all the unlinked port (or

"underway") emissions are summed within a county and evenly distributed to the shapes that are identified within
that county (either ports or shipping lanes).
The CMV sector has no data allowing for the designation of hourly time structure. Hence, the emissions are
temporally invariant over all hours of the year ("flat" distribution).
*Uncertainty*
The uncertainty of the CMV emissions is directly inherited from the uncertainty estimation described for the point
and nonpoint source processing, respectively. The only difference related to the the CO magnitude uncertainty
(±10.0%) which was derived from examination of the range of carbon content and fuel density uncertainties outlined
in EPA (2017), Annex 2, page A-87 for residual fuels, the dominant fuel used in CMV.
**2.1.10   Cement**
$CO_2$ is emitted from cement manufacturing as a result of fuel combustion and as process-derived emissions
(Andrew, 2018). The emissions from fuel combustion are captured in the point source reporting. The process-
derived $CO_2$ emissions result from the chemical process that converts limestone to calcium oxide and $CO_2$. This
occurs during "clinker" production (clinker is the raw material for cement which is produced by grinding the clinker
material).
Estimation of $CO_2$ emissions from clinker production utilizes two datasets. The first is the data provided by the
Portland Cement Association which provides the annual clinker capacity at individual facilities, postal addresses,
facility name, zip code and contact phone numbers (PCA, 2006). The capacity data reflects conditions for the
calendar year 2006. The other dataset utilized is the Minerals Yearbook produced by the United States Geological
Survey which provides the capacity factor (or percent utilization of capacity) on a statewide or multi-state basis
(some states are quantified individually, others are part of an aggregate) (USGS 2013). The product of capacity and
the capacity factor provides an estimate of clinker production.
Clinker production for 2011 is scaled from the Vulcan version 2.0 (CY 2002) estimate (Gurney et al., 2009) using
the relative annual capacity factor. The $CO_2$ emission factor used in the Vulcan Project is 0.59 metric tonnes
$CO_2$/short ton of clinker produced (IPCC, 2006).
The geolocation for each of the individual facilities was achieved by entering the PCA document's facility address
into Google Earth and visually inspecting the scene for the primary emitting stack of the cement facility. This
approach succeeded in locating all 105 facilities present in the PCA document.
The EPA estimates cement manufacturing in 2011 to account for 32.2 $MtCO_2$/year (USEPA 2017). These estimates,
in turn, are based upon throughput estimates from the U.S. Geological Survey. Vulcan estimates a total of 34.6
$MtCO_2$/year which compares well with the cement manufacturing estimate from the EPA.
The cement sector has no data allowing for the designation of hourly time structure. Hence, the emissions are evenly
distributed over all hours of the year (a "flat" distribution).



*Uncertainty*
The uncertainty in the cement emissions sector is currently prescribed as +/- 10% for the 95% CI. We use a
comparison between the facility-scale sum of clinker production in a state and the USGS state throughput (estimated
from the capacity factor and capacity). The mean percentage difference across all states and multistate aggregates
was 9.8%, which was rounded to 10% and intepreted as a 95% CI value.
**2.2   Multiyear estimation**
The multiyear (2010-2015) results were achieved using scale factors constructed from the EIA State Energy Data
System (SEDS) database (http://www.eia.gov/state/seds/)..Ratios were constructed relative to the year 2011 in all
SEDS sector/fuel designations for each US state. The crosswalk from the EIA SEDS codes to a sector/fuel
designation is provided in supplementary material, Table S8.
Exceptions to the use of the EIA SEDS database were made for the electricity production, railroad and CMV
multiyear scaling. Electricity production $FFCO_2$ emissions are monitored on an hourly basis for all the output
derived from the CAMD data (92.4% of the total electricity production emissions) and on a monthly basis for all of
the EIA reported data (6.2% of the total electricity production emissions). The remaining NEI reported electricity
production emissions (1.4% of the total electricity production emissions) use the EIA SEDS multiyear ratios.
In the case of the railroad sector, state-scale EIA specific to distillate fuel oil sales to the railroad sector was used
(http://www.eia.gov/dnav/pet/pet_cons_821dsta_a_epd0_val_mgal_a.htm) to construct the year-to-year ratios
relative to 2011. This data is used in generating the results in the EIA SEDS database but is aggregated and thus not
as specific to the railroad sector as needed. Large year-over-year ratio values were found for a few individual years
in low-population states (Nevada, Rhode Island, New Mexico, Hawaii). Values that exceeded 5.0 were replaced by
the year-specific US average ratio.
The procedure for the CMV $FFCO_2$ emissions is similar but combines the EIA data on distillate fuel oil sales for
"vessel bunkering use" (http://www.eia.gov/dnav/pet/pet_cons_821dsta_a_epd0_vab_mgal_a.htm) with residual
fuel oil sales for transportation (http://www.eia.gov/dnav/pet/pet_cons_821rsda_a_eppr_vat_mgal_a.htm). As with
the railroad sector application, large year-over-year ratios were filtered (those exceeding 5.0 were replaced by the
US national average).
The ratio values are applied to the annual totals in each of the sector/fuel categories specific to the state FIPS code to
generate a multiyear time series.
**3. Results**
Annual sector totals are provided in Table 6 for the 2010-2015 time period. Across all sectors, 2012 is the year with
the least emissions (1529.4 MtC/yr; 95% CI: 1249-1846 MtC/yr). While 2010 was the largest year for total
emissions (1638.2; 95% CI: 1338-1977 MtC/yr), the maximum value was primarily due to large $FFCO_2$ emissions in



the electricity production sector. The total FFCO$_2$ emissions (plus cement) in 2015, the most recent year in the time
series, were 1543.7 MtC/year (95% CI: 1268, 1857), a decline driven almost entirely by electricity production
FFCO$_2$ emissions. Electricity production is the largest emitting sector in all years, followed by the onroad and
industrial sectors, respectively.
**Table 6: Annual sector specific FFCO$_2$ (and cement) emission totals for the United States, 2010-2015,**
**estimated by Vulcan v3.0. (units: MtC/year)**

| Sector\Year | 2010 | 2011 | 2012 | 2013 | 2014 | 2015 |
|---|---|---|---|---|---|---|
| Residential | 92.0 | 89.2 | 78.5 | 91.3 | 95.3 | 88.0 |
| Commercial | 63.0 | 62.9 | 57.1 | 63.4 | 66.8 | 68.5 |
| Industrial | 230.6 | 228.4 | 227.2 | 233.8 | 237.4 | 231.4 |
| Elec Prod | 667.3 | 641.0 | 604.3 | 609.0 | 609.2 | 574.4 |
| Onroad | 452.0 | 440.6 | 436.6 | 443.1 | 448.6 | 452.4 |
| Nonroad | 63.6 | 62.7 | 61.8 | 62.9 | 64.0 | 64.6 |
| Airport | 19.8 | 19.6 | 20.5 | 22.3 | 22.3 | 21.8 |
| Rail | 11.9 | 11.9 | 12.6 | 13.7 | 15.1 | 14.6 |
| CMV | 28.4 | 23.3 | 20.9 | 18.5 | 16.2 | 18.1 |
| Cement | 9.5 | 9.7 | 9.8 | 9.8 | 9.8 | 9.8 |
| Total | 1638.2 (1338, 1977) | 1589.3 (1299, 1917) | 1529.4 (1249,1846) | 1567.9 (1284,1889) | 1584.6 (1300,1908) | 1543.7 (1268,1857) |

The order of the 2011 FFCO$_2$ emitting sectors (Figure 3) varies regionally (US Census Regions) with the electricity
production sector accounting for the largest share in the Midwest (44%) and South (46%) while onroad emissions
account for the largest share in the West (32%) and Northeast (29%). The sum of the commercial and residential
sectors are a larger share of total emissions in the Northeast (22%) than in the other three regions (6%-11%). The
industrial FFCO$_2$ emissions account for the largest industrial share in the West (19%) compared to the other three
regions (13%-14%). Overall, 2011 FFCO$_2$ emissions are largest in the South (652 TgC), followed by the Midwest
(434 TgC), the West (293 TgC) and the Northeast (200 TgC).

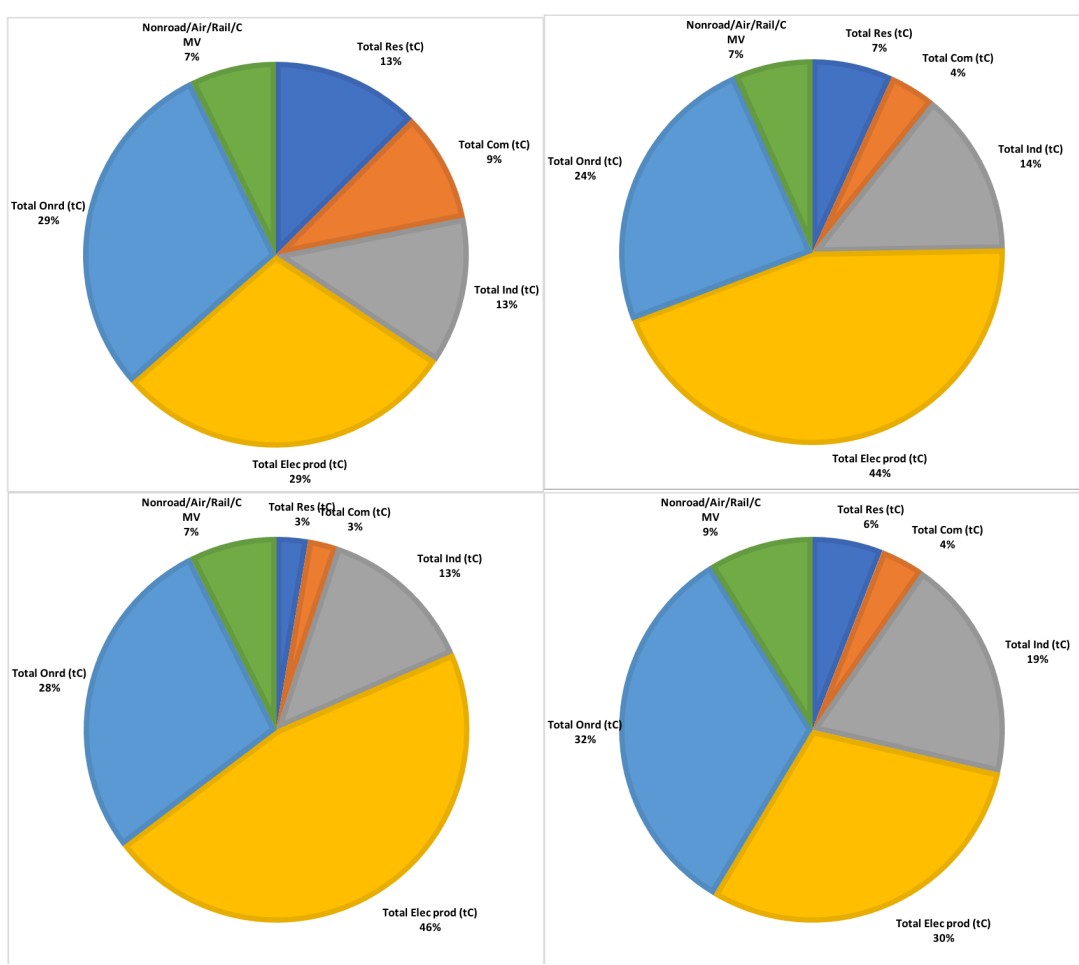

**Figure 3: Sector-specific percentage share of 2011 Vulcan v3.0 FFCO₂ emissions for the United States by US Census Region: a) Northeast; b) Midwest; c) South; d) West.**

When examined at the state-scale, the apportioning of the FFCO₂ emitting sectors shows a relationship to total

FFCO₂ per capita emissions (Figure 4). States with larger per capita emissions tend to be dominated by industrial

and electricity production sector FFCO₂ emissions. States with lower per capita total FFCO₂ emissions tend to have

lesser industrial and electricity production FFCO₂ emissions and a greater share of onroad and

residential/commercial emissions. A few states are notable exceptions to this pattern. For example, the states of

Alaska, Washington, and South Dakota have a relatively large portion of nonroad emissions while Rhode Island and

Washington DC have a relatively large proportion of commercial sector FFCO₂ emissions. Tabular results at the

state-scale are provided in supplementary material, Table S9.

Per capita emissions vary across the states, with the largest in the state of Wyoming (38.5 tC/person) and the

smallest in Washington DC (2.11 tC/person) and California (2.81 tC/person). The median total per capita FFCO₂

emissions at the county-scale are 3.80 tC/person (see Figure S3 in supplementary material). It is worth noting that

the population statistics used here define the population as that residing within the state which will influence the
results for Washington DC where there is a large daytime non-resident population.

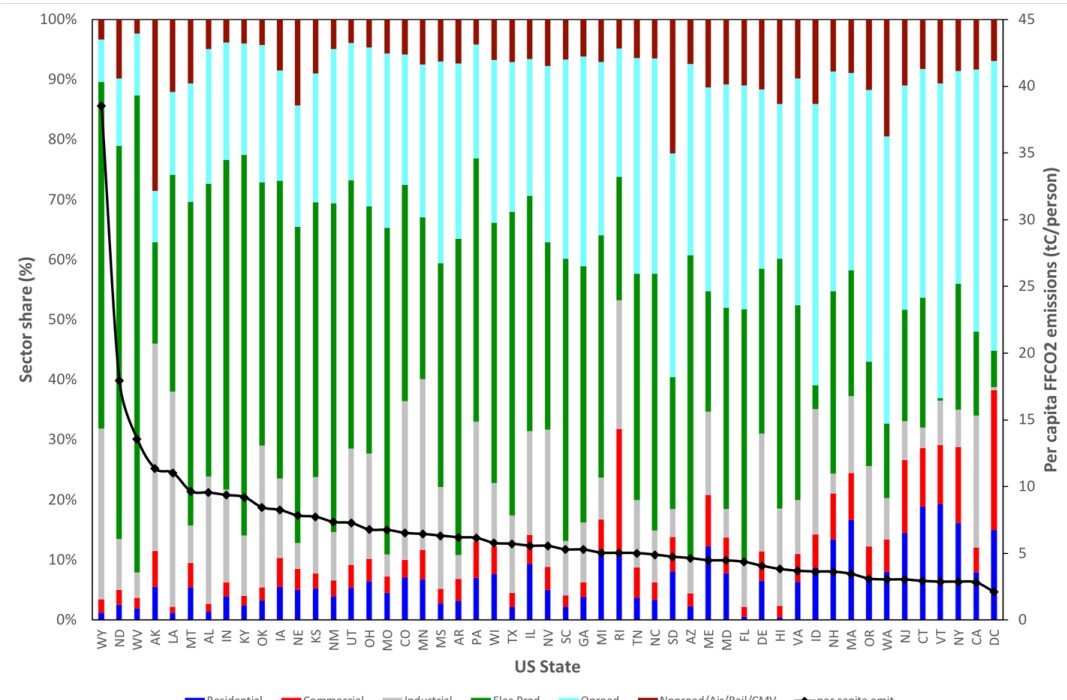

**Figure 4: Vulcan v3.0 FFCO₂ emissions sector share (left y-axis: %) by state and per capita FFCO₂ emissions (right y-axis: tC/person) for year 2011.**

The Vulcan FFCO₂ emissions are quantified at the sub-national scale according to three general shape types: points
(e.g. electricity production, industrial point reporting), lines (e.g. onroad) and polygons (e.g. nonroad, residential).
For use in atmospheric transport modeling and ease of use in analysis, these results are gridded using a 1km x 1km
regular grid (Figure 5a). The importance of urban areas is clearly demonstrated in the complete US mapped
landscape along with the greater urbanization in the eastern half of the country and along the West coast. Interstates
and other large primary roadways are also evident across the US connecting large population centers. Normalization
by population (Landscan, 2017) offers a dramatically different perspective on U.S. FFCO₂ emissions, placing
greater emphasis on the western half of the country (Figure 5b).

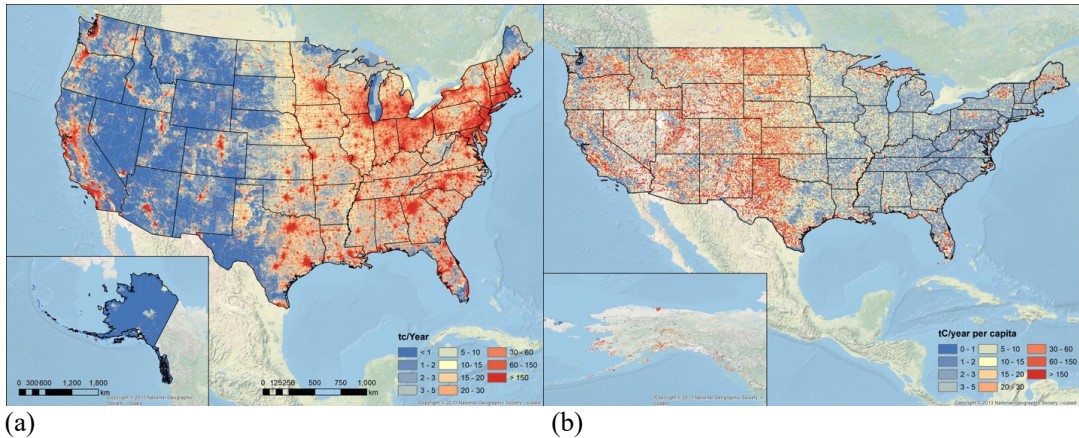

(a)                                                    (b)
**Figure 5: Vulcan v3.0 2011 FFCO$_2$ emissions for the United States. a) absolute emissions (1km x 1km**
**resolution, tC); b) per capita emissions (0.1° x 0.1° resolution, tC; different resolution and projection required**
**for integration with population data). Copyright © National Geographic Society, i-cubed.**
A center of mass (CoM) is a useful and compact metric to understand and illustrate the spatial changes in fossil fuel
CO$_2$ emissions over time (Gregg et al., 2009). The CoM summarizes the distribution of emissions in the same way
as the mean summarizes a probability distribution (Asefi et al., 2014). Figure 6 shows both the multiyear and
monthly mean CoM. The multiyear CoM shows a general shift from the East to the West over the six years
examined here with the CoM located in Missouri approximately 70 miles SW of St Louis, MO. The monthly mean
results show a tendency to move along a NE/SW axis with wintertime movement towards the NE driven by greater
heating needs associated with cold/continental conditions. Summertime movement is towards the SW associated
with the rising air-conditioning demand during summer months. The months of May/June/July show movement
towards the SE in May and June, a shift towards the North in July, before resuming the Western shift in August and
September.
The 2011 monthly FFCO$_2$ emissions magnitude exhibits two maxima of roughly equal value over the course of the
year: a winter maximum in the months of December and January and a summer maximum in the months of July and
August. The maxima correspond to the northermost CoM position in the winter and near-southernmost CoM
position in summer which are associated with the demand for heating in the winter, dominated by more northerly
locations, and the demand for cooling in the summer, dominated by more southerly locations.

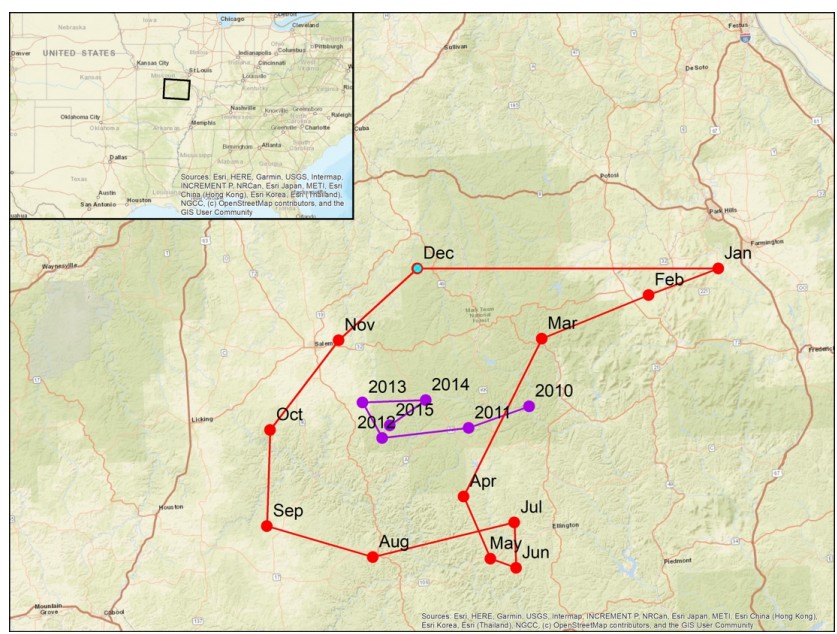

**Figure 6: Vulcan v3.0 FFCO$_2$ emissions center of mass estimate. Red line/symbols denote 2010-2015 annual time series. Purple line/symbols denote monthly mean FFCO$_2$ emissions. This map was made in ArcMap$^{TM}$ by Esri using the OpenStreetMap basemap layer (Copyright © Esri, with data from OpenStreetMap contributers ©).**

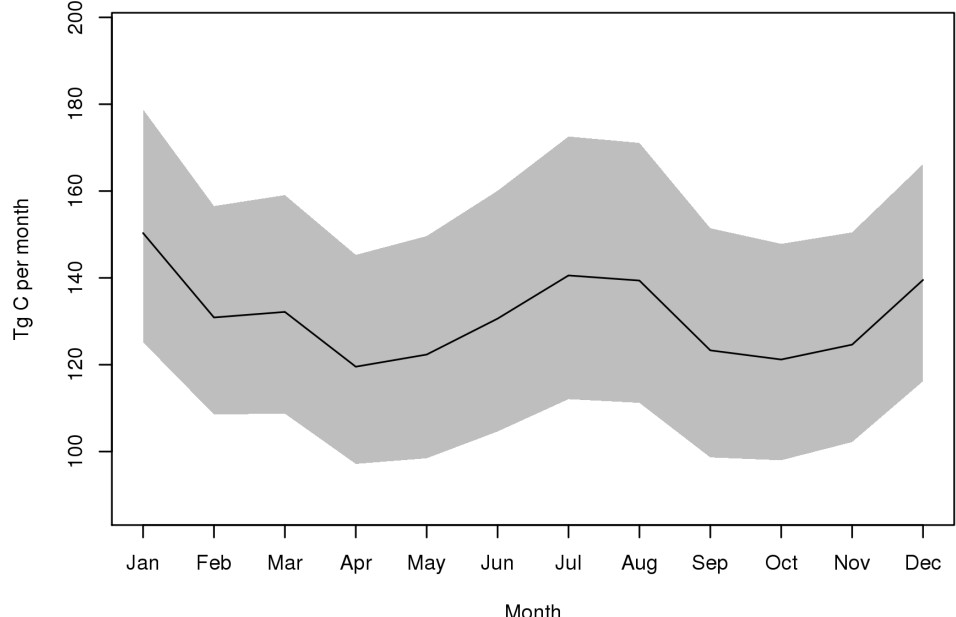

**Figure 7. Vulcan v3.0 2011 FFCO$_2$ emissions for the United States by month with 95% confidence interval. Units: TgC/month.**



## 4. Discussion

The Vulcan approach to quantification of bottom-up granular $FFCO_2$ emissions established a method that has been since followed by other investigators with useful and instructive variations (e.g. Bun et al., 2019; Gately et al., 2017). Some of the differences are driven by differing national circumstances related to data availability and collection sources. However, other than the ACES data product, which covers only the NorthEast US domain, there is no other US-based granular estimate of $FFCO_2$ emissions with which to evaluate the results presented here. As noted in the introduction, however, numerous global gridded estimates of $FFCO_2$ emissions have been constructed starting in the 1990s. Currently only the ODIAC estimate is quantified at the same 1km x 1km resolution as found in Vulcan. Hence, we perform comparison to the ODIAC output over the Vulcan domain in the hope of providing insight into one or both of the emission estimates. We masked the ODIAC output with a mask that includes all land surface gridcells and all gridcells offshore for which Vulcan possesses a non-zero emission value. We estimate the ODIAC emissions to be 1453.7 TgC/year for the year 2011. The same mask applied to Vulcan results in $FFCO_2$ emissions of 1553.8 TgC/yr or a difference of 100.1 MtC/yr (7.6%). We also removed all CMV emissions from Vulcan due to the fact that the ODIAC 1km x 1km data product does not include any bunker fuels in the emissions. We make no adjustment to the Vulcan airport emissions, though a portion is also likely in the bunker fuel category. The inability to precisely isolate the bunker fuel amounts from Vulcan will result in comparison uncertainties but these are considered small relative to the scale at which the comparison is made.

At the individual gridcell spatial scale, further detail on differences between the two data products can be examined (Figure 8). Three different relationships appear in the spatial gridcell comparison with an correlation coefficient of 0.69 and a slope of 0.43 (change in ODIAC/change in Vulcan). The first shows good correlation close to the 1:1 line for large emitting gridcells. These are gridcells dominated by power production facilities and hence, traced to common regulatory data reporting in the two data products. The second relationship evident in the paired gridcell comparison shows rough correspondence whereby Vulcan has a larger range of emission values to a narrower ODIAC range rotated clockwise from the 1:1 line. There is also a well-defined lower threshold of emissions in ODIAC (~11 tC/yr), likely tied to the threshold associated with low levels of nighttime lighting, a dominant driver of the ODIAC spatial distribution (Liang et al., 2019). The third discrete relationship is a non-correlated collection of paired gridcells in the upper range of ODIAC emissions for which the Vulcan counterparts exhibit midrange emission values.

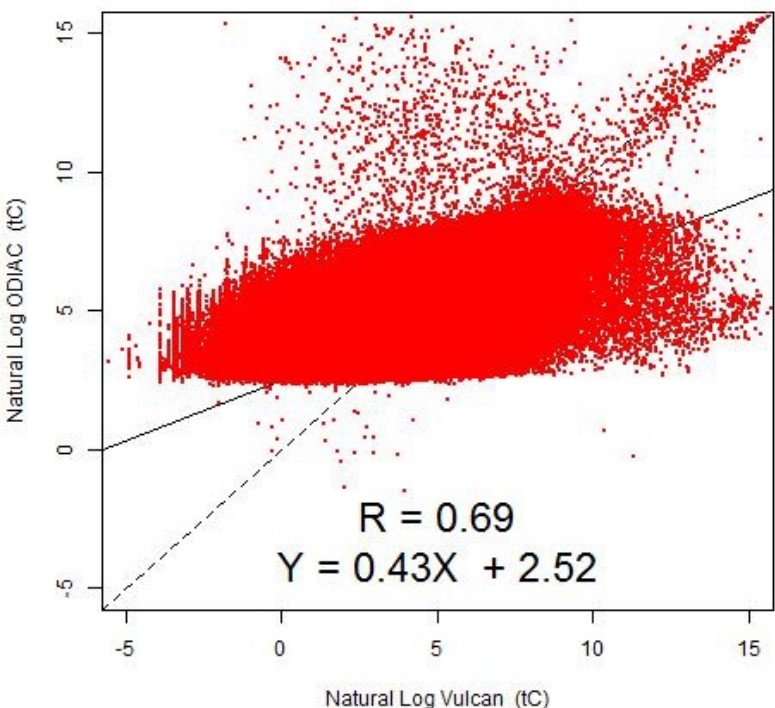

**Figure 8. Comparison of log-transformed ODIAC (y-axis) and Vulcan v3.0 (x-axis) FFCO₂ emissions (units: Natural log tC).**

When presented explicitly in space, total ODIAC and Vulcan FFCO₂ emissions show similar spatial patterns at the
domain-wide scale, characterized by large concentrations in urban centers across the US landscape, particularly
along the Northeastern seaboard and the upper Midwest (Figure 9a, 9c). ODIAC exhibits large numbers of gridcells
in rural areas across the Western U.S. with no emission value, likely due to the lack of a nighttime light signal in
those areas. This is further demonstrated by the emission histogram (Figure 9b, 9d) whereby ODIAC has a distinct
lower cutoff at 11.02 tC/yr (natural log of which is 2.4) compared to Vulcan which has a more continuously
declining low value distribution. The maximum emission frequency bin for ODIAC is centered at 23.3 tC/yr
whereas the equivalent value for Vulcan is 12.8 tC/yr. Vulcan gridcells in these areas have emission values but they
are small in comparison to more populated areas and can be dominated by nonroad emissions which use large spatial
proxies for distribution. In estimating the gridcell-scale relative emissions difference (GRD), these pairs are
excluded. GRD values are high throughout the populated portions of the U.S., particularly in the Eastern half of the
country. There are large spatially continuous areas in which ODIAC emissions exceed Vulcan and vice-versa. Large
differences occur in urban centers, most notably in the Western U.S., with Vulcan often exceeding ODIAC
emissions in the urban core but ODIAC exceeding Vulcan outside of the urban core in these cities. (e.g. Phoenix,
Dallas, Los Angeles, St Louis).





To provide an average relative difference between the two data products, we calculate the gridcell absolute median
relative difference, *GAMRD*, the median of a set of individual paired gridcell relative differences, where the
differences are represented in absolute units (i.e. so all GRD values are positive). GAMRD is calculated as,

$$GAMRD = med \left\{ \frac{abs\left(E_i^A - E_i^V\right)}{\frac{E_i^A \mp E_i^V}{2}} \right\}$$
(1)

where $E$ represented emissions for the ODIAC ($A$) and Vulcan ($V$) for each $i^{th}$ paired gridcell. We only include
gridcell pairs in which neither of the emission values is zero. We find that the GAMRD between the ODIAC and
Vulcan v3.0 $FFCO_2$ emissions at the 1 km x 1 km spatial scale is 80.04%.

e                                            f

**Figure 9. Comparison of the ODIAC and Vulcan total FFCO₂ emissions as mapped distributions (left) and
frequency histograms (right) for contiguous U.S. only: (a,b) ODIAC total FFCO₂ emissions; (c,d) Vulcan total
FFCO₂ emissions; (e,f) GRD values ({ODIAC-Vulcan}/Vulcan) (Values larger than 99% and smaller than -
99% were excluded from the GRD frequency distribution).**



The other means by which to assess the Vulcan results is via comparison to recent work using $14CO_2$ measurements
and an atmospheric inversion approach by Basu et al. (2019). In that study, the mean of the ensemble of atmospheric
$CO_2$ inversion estimates was within 1.4% of the total US Vulcan estimate in the year 2011.
The increased resolution of the Vulcan v3.0 $FFCO_2$ emissions data product (1 km x 1 km) in comparison to the
previous 10 km x 10 km Vulcan v2.0 data product raises the prospect of supplying information that resolves sub-city
spatial scales (e.g. neighborhood) in a comprehensive fashion across the entire U.S. landscape. At this resolution,
most urbanized areas in the U.S. would comprise domains much larger than the 1 km x 1 km resolution and, hence,
have sub-domain information emissions content. In this way, the Vulcan v3.0 $FFCO_2$ emissions data product offers a
scope 1, high-resolution inventory estimate for every urbanized area in the U.S. (Figure 10).

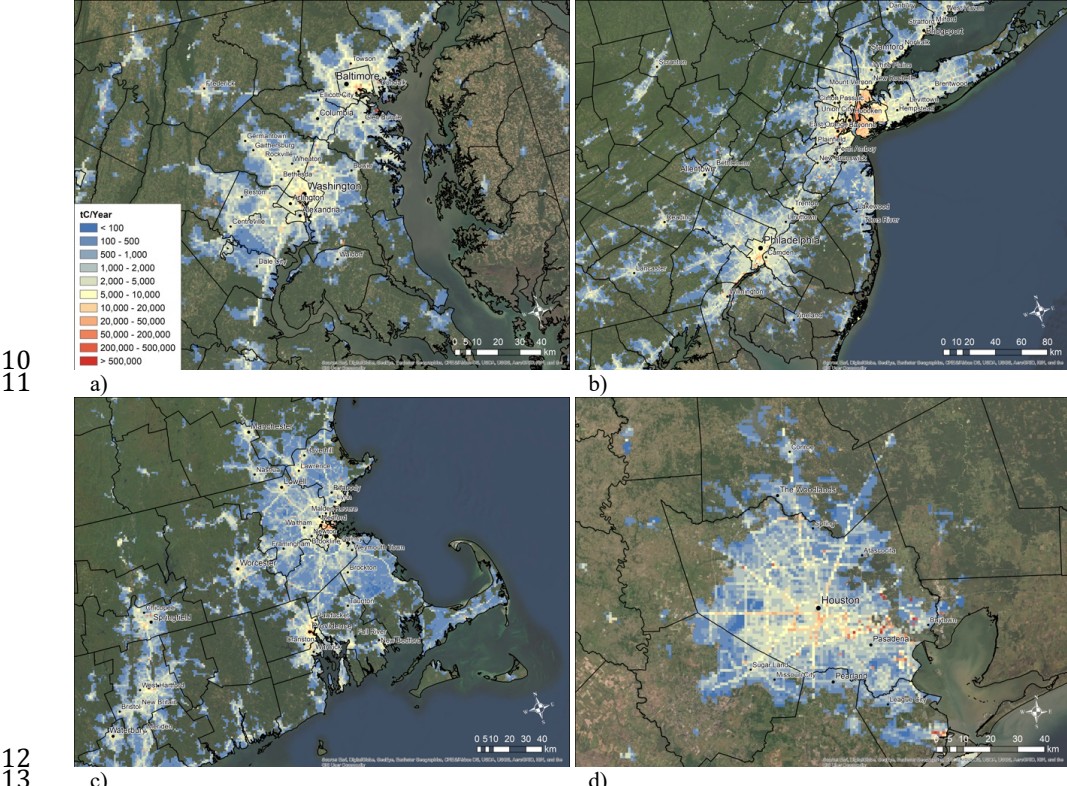

a)                                          b)
c)                                          d)

Open Access    Earth System
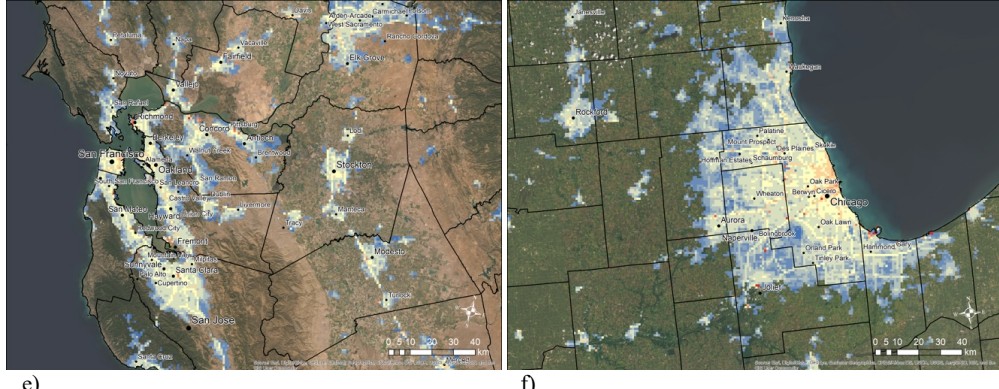

e)                                                           f)
**Figure 10: Vulcan v3.0 FFCO$_2$ emissions for a selected group of U.S. urban areas. a) Washington DC; b) New**
**York; c) Boston; d) Houston; e) San Francisco; f) Chicago. These maps were made in ArcMap$^{TM}$ by Esri**
**using the World Imagery basemap layer (Copyright © Esri).**
After adopting the US census "urbanized area" boundary definition (https://www.census.gov/programs-
surveys/geography/guidance/geo-areas/urban-rural/2010-urban-rural.html), total US FFCO$_2$ emissions within these
urban area boundaries come to 45.1% of the total Vulcan FFCO$_2$ contiguous US emissions in 2011 (709.6 TgC). We
narrow urban emissions to the sum of residential, commercial and onroad in an effort to eliminate emissions sectors
that are often historically artifactual to location within a given urban area (e.g. power plants, industrial facilities) and
hence, less directly related to urban residents and their emitting activities. The sum of these three sectors within the
urbanized area boundary accounts for 65% of the these same three sectors at the national scale, slightly less than the
proportion of the US population within the urbanized area boundary, 73%. This indicates that for the sum of the
residential, commercial and onroad sectors, urban residents emit less per capita than non-urban residents in the
contiguous US.
**5 Data availability, policy and future updates**
The sector-specific Vulcan v3.0 annual gridded emissions data product can be downloaded from the Oak Ridge
National Laboratory Distributed Active Archive Center (ORNL DAAC)
(https://doi.org/10.3334/ORNLDAAC/1741) and is distributed under Creative Commons Attribution 4.0
International (CC-BY 4.0, https://creativecommons.org/licenses/by/ 4.0/deed.en). The Vulcan v3.0 FFCO$_2$
emissions data product is provided in annual 1 km x 1 km NetCDF file formats, one file for each of the 6 years
(2010-2015). The annual files vary in size (by sector) with the largest individual file being approximately 50 GB.
Separate files are available for Alaska and the contiguous United States (Gurney et al., 2019).
Attempts will be made to update the Vulcan FFCO$_2$ emissions on a roughly bi-annual basis, depending upon support
and the availability of data sources described in this study.





**6 Conclusions**
Fossil fuel carbon dioxide ($FFCO_2$) emissions, spanning the years 2010-2015, at a spatial scale of 1km x 1km and an
hourly temporal scale have been completed for the United States under the Vulcan Project. These Vulcan version 3.0
emissions are constructed through a bottom-up approach, gathering and combining data from multiple sources such
as CO emissions reporting, direct flux measurements, and traffic monitoring. We describe the complete Vulcan
version 3.0 methodology here, sector-by-sector in addition to the methods for uncertainty estimation.
We estimate $FFCO_2$ emissions for the year 2011 of 1589.3 TgC with a 95% confidence interval of 1299/1917 TgC
(+18.3%/-20.6%), implying a one-sigma uncertainty of ~ ±10%. The order of the 2011 $FFCO_2$ emitting sectors
shows the electricity production sector accounting for the largest share in the Midwest (44%) and South (46%) while
onroad emissions account for the largest share in the West (32%) and Northeast (29%). Overall, 2011 $FFCO_2$
emissions are largest in the South (652 TgC), followed by the Midwest (434 TgC), the West (293 TgC) and the
Northeast (200 TgC).
We find that per capita $FFCO_2$ emissions are larger in states proportionately dominated by the electricity production
and industrial sectors and smaller in states proportionately dominated by onroad and residential/commercial building
emissions. The center of mass (CoM) of $FFCO_2$ emissions in the US are located in the state of Missouri with mean
seasonality that extends towards the Northeast in the wintertime then moves towards the Southwest in the summer,
likely reflecting the seasonal demand for heating and air-conditioning. Comparison to ODIAC, a global gridded
$FFCO_2$ emissions estimate shows large differences in both total emissions (100.1 TgC for year 2011) and spatial
patterns. The spatial correlation ($R^2$) between the two data products was 0.38 and the mean absolute difference at the
individual gridcell scale was 80.04%.
The Vulcan v3.0 $FFCO_2$ emissions data product offers an immediate high-resolution estimate of emissions in every
city within the U.S., providing a large potential savings of time and effort for cities planning to develop self-reported
city inventories. Research associated with comparison to existing self-reported urban inventories and the addition of
indirect $FFCO_2$ emissions (scope 2 & 3) are underway.
**Author Contributions.** KRG conceived of the content, performed portions of the data collection, coding and
analysis, and wrote the paper. JL, RP, YS, and JH performed portions of the data collection, coding and analysis. GR
performed analysis.
**Competing Interests.** The authors declare that they have no conflict of interest.
**Financial Support.** This research was made possible through support from the National Aeronautics and Space
Administration grant NNX14AJ20G and the NASA Carbon Monitoring System program, Understanding User
Needs for Carbon Information project (subcontract 1491755).



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

Methods to Support International Climate Agreements. *Greenhouse Gas Measurement and Management*, *1*(2), 132–
133. https://doi.org/10.1080/20430779.2011.579358, 2013.
Jessberger, Steven: personal communication: Engineer, Federal Highway Administration, Office of Highway Policy
Information, Travel Monitoring and Surveys, 1200 New Jersey Avenue, S.E., HPPI-30, E83-418, Washington, DC
20590, USA, 202-366-5052, 202-366-7742), 2016.
Jones, C. M., & Kammen, D. M.: Quantifying carbon footprint reduction opportunities for U.S. households and
communities. *Environmental Science and Technology*. https://doi.org/10.1021/es102221h, 2011.
Jones, C., & Kammen, D. M.: Spatial Distribution of U.S. Household Carbon Footprints Reveals Suburbanization
Undermines Greenhouse Gas Bene fi ts of Urban Population Density. *Environmental Science & Technology*, *48*(2),
895–902. https://doi.org/10.1021/es4034364, 2014.
Kurokawa, J., Ohara, T., Morikawa, T., Hanayama, S., Janssens-Maenhout, G., Fukui, T., … Akimoto, H.:
Emissions of air pollutants and greenhouse gases over Asian regions during 2000-2008: Regional Emission
inventory in ASia (REAS) version 2. *Atmospheric Chemistry and Physics*, *13*(21), 11019–11058.
https://doi.org/10.5194/acp-13-11019-2013, 2013.
This product was made utilizing the LandScan (2017)™ High Resolution global Population Data Set copyrighted by
UT-Battelle, LLC, operator of Oak Ridge National Laboratory under Contract No. DE-AC05-00OR22725 with the
United States Department of Energy. The United States Government has certain rights in this Data Set.
LeQuéré, C., Andrew, R., Friedlingstein, P., Sitch, S., Hauck, J., Pongratz, J., … Zheng, B.: Global Carbon Budget
2018. *Earth System Science Data*. https://doi.org/10.5194/essd-10-2141-2018, 2018.
Liu, Z., Bambha, R. P., Pinto, J. P., Zeng, T., Boylan, J., Huang, M., … Michelsen, H. A.: Toward verifying fossil
fuel $CO_2$ emissions with the CMAQ model: Motivation, model description and initial simulation. *Journal of the Air
and Waste Management Association*, *64*(4), 419–435. https://doi.org/10.1080/10962247.2013.816642, 2014.
Liu, F., Zhang, Q., Tong, D., Zheng, B., Li, M., Huo, H., & He, K. B.: High-resolution inventory of technologies,
activities, and emissions of coal-fired power plants in China from 1990 to 2010. *Atmospheric Chemistry and
Physics*, *15*(23), 13299–13317. https://doi.org/10.5194/acp-15-13299-2015, 2015.
Macknick, J.: Energy and $CO_2$ emission data uncertainties. *Carbon Management*. Future Science LtdLondon, UK.
https://doi.org/10.4155/cmt.11.10, 2011.
Manufacturing Energy Consumption Survey (2010) 2010 MECS Survey Data, U.S. Energy Information
Administration. Retrieved from: https://www.eia.gov/consumption/manufacturing/data/2010/#r10 (Aug 1, 2018).
Marland, G., Rotty, R. M., & Treat, N. L.: $CO_2$ from fossil fuel burning: global distribution of emissions. *Tellus B*,
*37 B*(4–5), 243–258. https://doi.org/10.1111/j.1600-0889.1985.tb00073.x, 1985.
Mendoza, D., Gurney, K. R., Geethakumar, S., Chandrasekaran, V., Zhou, Y., & Razlivanov, I.: Implications of
uncertainty on regional CO2 mitigation policies for the U.S. onroad sector based on a high-resolution emissions
estimate. Energy Policy, 55, 386–395. https://doi.org/ 10.1016/j.enpol.2012.12.027., 2013.
Minx, J., Baiocchi, G., Wiedmann, T., Barrett, J., Creutzig, F., Feng, K., … Hubacek, K.: Carbon footprints of cities
and other human settlements in the UK. *Environmental Research Letters*, *8*(3). https://doi.org/10.1088/1748-
38   9326/8/3/035039, 2013.

Moran, D., Kanemoto, K., Jiborn, M., Wood, R., Többen, J., & Seto, K. C.: Carbon footprints of 13 000 cities.
*Environmental Research Letters*, *13*(6). https://doi.org/10.1088/1748-9326/aac72a, 2018.
National Research Council: *Verifying Greenhouse Gas Emissions: Methods to Support International Climate
Agreements*. Washington DC: The National Academies Press. https://doi.org/10.17226/12883, 2010.
Oda, T., & Maksyutov, S.:. A very high-resolution (1km×1 km) global fossil fuel CO2 emission inventory derived
using a point source database and satellite observations of nighttime lights. *Atmospheric Chemistry and Physics*.
https://doi.org/10.5194/acp-11-543-2011, 2011.



Oda, T., Maksyutov, S., & Andres, R. J.:. The Open-source Data Inventory for Anthropogenic $CO_2$, version 2016 (ODIAC2016): A global monthly fossil fuel CO2 gridded emissions data product for tracer transport simulations and surface flux inversions. *Earth System Science Data*. https://doi.org/10.5194/essd-10-87-2018, 2018.

Ohara, T., Akimoto, H., Kurokawa, J., Horii, N., Yamaji, K., Yan, X., … Asian, A.: An Asian emission inventory of anthropogenic emission sources for the period 1980 ? 2020 To cite this version : and Physics An Asian emission inventory of anthropogenic emission sources for the period 1980 – 2020. *Atmospheric Chemistry and Physics, 7*((16)), 4419–4444, 2007.

Olivier, J. G. J., Bloos, J. P. J., Berdowski, J. J. M., Visschedijk, A. J. H., & Bouwman, A. F.: A 1990 global emission inventory of anthropogenic sources of carbon monoxide on $1° \times 1°$ developed in the framework of EDGAR/GEIA. *Chemosphere - Global Change Science*, *1*(1–3), 1–17. https://doi.org/10.1016/S1465-9972(99)00019-7, 1999.

Olivier, J. G. J., Van Aardenne, J. A., Dentener, F. J., Pagliari, V., Ganzeveld, L. N., & Peters, J. A. H. W.: Recent trends in global greenhouse gas emissions:regional trends 1970–2000 and spatial distributionof key sources in 2000. *Environmental Sciences*, *2*(2–3), 81–99. https://doi.org/10.1080/15693430500400345, 2005.

Ou, J., Liu, X., Li, X., Li, M., & Li, W.: Evaluation of NPP-VIIRS nighttime light data for mapping global fossil fuel combustion CO2 emissions: A comparison with DMSP-OLS nighttime light data. *PLoS ONE*, *10*(9), e0138310. https://doi.org/10.1371/journal.pone.0138310, 2015.

Patarasuk, R., Gurney, K. R., O'Keeffe, D., Song, Y., Huang, J., Rao, P., … Ehleringer, J. R.: Urban high-resolution fossil fuel CO2 emissions quantification and exploration of emission drivers for potential policy applications. *Urban Ecosystems*, *19*(3), 1013–1039. https://doi.org/10.1007/s11252-016-0553-1, 2016.

Peylin, P., Houweling, S., Krol, M. C., Karstens, U., Rödenbeck, C., Geels, C., … Heimann, M.: Importance of fossil fuel emission uncertainties over Europe for CO2 modeling: Model intercomparison. *Atmospheric Chemistry and Physics*, *11*(13), 6607–6622. https://doi.org/10.5194/acp-11-6607-2011, 2011.

Petron, G, Tans, P., Frost, G., Chao, D., and Trainer, M.: High resolution emissions of $CO_2$ from power generation in the USA, *J. Geophys. Res.,* 113 doi:10.1029/2007/JG000602, 2008.

Pincetl, S., Chester, M., Circella, G., Fraser, A., Mini, C., Murphy, S., … Sivaraman, D.: Enabling Future Sustainability Transitions: An Urban Metabolism Approach to Los Angeles Pincetl et al. Enabling Future Sustainability Transitions. *Journal of Industrial Ecology*, *18*(6), 871–882. https://doi.org/10.1111/jiec.12144, 2014.

Portland Cement Company, Economic Research Department: *U.S. and Canadian Portland Cement Industry Plant Information Summary*, Portland Cement Association, Skokie, IL, 2006.

Quick, J. C.: Carbon dioxide emission factors for U.S. coal by origin and destination. *Environmental Science and Technology*, *44*(7), 2709–2714. https://doi.org/10.1021/es9027259, 2010.

Rayner, P. J., Raupach, M. R., Paget, M., Peylin, P., & Koffi, E.: A new global gridded data set of $CO_2$ emissions from fossil fuel combustion: Methodology and evaluation. *Journal of Geophysical Research Atmospheres*, *115*(19), 1–11. https://doi.org/10.1029/2009JD013439, 2010.

Residential Energy Consumption Survey (2013) 2009 RECS Survey Data, U.S. Energy Information Administration. Retrieved from: https://www.eia.gov/consumption/residential/data/2009/index.php?view=microdata (Aug 1, 2018).

Shu, Y., & Lam Nina, N. S. N. (2011). Spatial disaggregation of carbon dioxide emissions from road traffic based on multiple linear regression model. *Atmospheric Environment*, *45*(3), 634–640. https://doi.org/10.1016/j.atmosenv.2010.10.037.

United States Environmental Protection Agency (1995) *FIRE Version 5.0 Source Classification Codes and Emission Factor Listing For Criteria Air Pollutants*, Office of Air Quality Planning and Standards, Research Triangle Park, NC 27711, EPA-454/R-95-012, August 1995.

United States Environmental Protection Agency (2005a), *Emissions Inventory Guidance for Implementation of Ozone and Particulate Matter National Ambient Air Quality Standards (NAAQS and Regional Haze Regulations)*, Emissions Inventory Group, Emissions, Monitoring and Analysis Division, Office of Air Quality Planning and Standards, U.S. Environmental Protection Agency, Research Triangle Park, NC 27711, EPA-454/R-05-001, August.



United States Environmental Protection Agency (2005b), *Plain English Guide to the Part 75 Rule*, U.S. Environmental Protection Agency, Clean Air Markets Division.

United States Environmental Protection Agency (2005c), *EPA's National Mobile Inventory Model (NMIM), A consolidated emissions modeling system for MOBILE6 and NONROAD*, Office of Transportation and Air Quality, Assessment and Standards Division, U.S. Environmental Protection Agency, EPA420-R-05-024, December.

United States Environmental Protection Agency (2005d), *User's Guide for the Final NONROAD2005 Model, Assessment and Standards,* Division Office of Transportation and Air Quality U.S. Environmental Protection Agency, December.

United States Environmental Protection Agency (2010) *Draft Part 75 Emissions Monitoring Policy Manual*, U.S. Environmental Protection Agency, Clean Air Markets Division, Washington, D.C., April 2010.

United States Environmental Protection Agency (2011) *2011 National Emissions Inventory, version 1 Technical Support Document*, June 2014 – Draft. Office of Air Quality Planning and Standards. Retrieved from: https://www.epa.gov/air-emissions-inventories/2011-national-emissions-inventory-nei-technical-support-document (August 12, 2018).

U.S. Environmental Protection Agency (2012) *Motor Vehicle Emission Simulator (MOVES): User Guide for MOVES2010b* Office of Transportation and Air Quality, EPA-420-B-12-001b. Retrieved from: https://nepis.epa.gov/Exe/ZyPDF.cgi?Dockey=P100EP28.pdf (August 12 , 2018).

United States Environmental Protection Agency (2015a) *2011 National Emissions Inventory, version 2 Technical Support Document*, U.S. Environmental Protection Agency, Office of Air Quality Planning and Standards, Air Quality Assessment Division, Emissions Inventory and Analysis Group, Research Triangle Park, North Carolina, August 2015. www.epa.gov/air-emissions-inventories/2011-national-emissions-inventory-nei-data

United States Environmental Protection Agency (2015b), 40 DFR Part 60, EPA-HQ-OAR-2013-0602; FRL-XXXX-XX-OAR, RIN 2060-AR33, Carbon Pollution Emission Guidelines for Existing Stationary Sources: Electric Utility Generating Units, August 3, 2015.

United States Environmental Protection Agency (2015c) *Technical Support Document (TSD) Preparation of Emissions Inventories for the Version 6.2, 2011 Emissions Modeling Platform*, U.S. Environmental Protection Agency, Office of Air Quality Planning and Standards, Air Quality Assessment Division, contacts: Alison Eyth, Jeff Vukovich, August 2015. Retrieved from https://www.epa.gov/air-emissions-modeling/2011-version-62-technical-support-document (July 27, 2018).

United States Environmental Protection Agency (2017) *Inventory of U.S. Greenhouse Gas Emissions and Sinks 1990-2015*, EPA 430-P-17-001.

United States Environmental Protection Agency (2018) *Inventory of U.S. Greenhouse Gas Emissions and Sinks 1990-2016*, EPA 430-R-18-003.

United States Geological Survey (2013) *2011 Minerals Yearbook: Cement,* U.S. Department of the Interior, U.S. Geological Survey. December 2013.

USGCRP, 2018: Second State of the Carbon Cycle Report (SOCCR2): A Sustained Assessment Report [Cavallaro, N., G. Shrestha, R. Birdsey, M. A. Mayes, R. G. Najjar, S. C. Reed, P. Romero-Lankao, and Z. Zhu (eds.)]. U.S. Global Change Research Program, Washington, DC, USA, 878 pp., https://doi.org/10.7930/SOCCR2.2018.

VandeWeghe, J. R., & Kennedy, C.: A Spatial Analysis of Residential Greenhouse Gas Emissions in the Toronto Census Metropolitan Area. *Journal of Industrial Ecology*, *11*(2), 133–144. https://doi.org/10.1162/jie.2007.1220, 2007.

Wang, R., Tao, S., Ciais, P., Shen, H. Z., Huang, Y., Chen, H., … Piao, S. L.: High-resolution mapping of combustion processes and implications for $CO_2$ emissions. *Atmospheric Chemistry and Physics*, *13*(10), 5189–5203. https://doi.org/10.5194/acp-13-5189-2013, 2013.

Wang, J., Cai, B., Zhang, L., Cao, D., Liu, L., Zhou, Y., … Xue, W.: High resolution carbon dioxide emission gridded data for China derived from point sources. *Environmental Science and Technology*, *48*(12), 7085–7093. https://doi.org/10.1021/es405369r, 2014.



Wilson, J., Spinney, J., Millward, H., Scott, D., Hayden, A., & Tyedmers, P.: Blame the exurbs, not the suburbs:
Exploring the distribution of greenhouse gas emissions within a city region. *Energy Policy*, *62*, 1329–1335.
https://doi.org/10.1016/j.enpol.2013.07.012, 2013.
World Business Council for Sustainable Development: *CO₂ accounting and reporting standard for the cement
industry*, version 2.0, June 2005.
Yadav, V., Michalak, A. M., Ray, J., & Shiga, Y. P.: A statistical approach for isolating fossil fuel emissions in
atmospheric inverse problems. *Journal of Geophysical Research*, *121*(20), 12,490-12,504.
https://doi.org/10.1002/2016JD025642, 2016.
Zhang, Y., Wang, H., Liang, S., Xu, M., Liu, W., Li, S., … Bi, J.: Temporal and spatial variations in consumption-
based carbon dioxide emissions in China. *Renewable and Sustainable Energy Reviews*, *40*, 60–68.
https://doi.org/10.1016/j.rser.2014.07.178, 2014.
Zheng, B., Huo, H., Zhang, Q., Yao, Z. L., Wang, X. T., Yang, X. F., … He, K. B.: High-resolution mapping of
vehicle emissions in China in 2008. *Atmospheric Chemistry and Physics*, *14*(18), 9787–9805.
https://doi.org/10.5194/acp-14-9787-2014, 2014.
Zhou, Y., and Gurney, K.R.:, Spatial relationships of sector-specific fossil fuel $CO_2$ emissions in the United States,
*Glob. Biogeochem. Cycles*, 25, GB3002, doi:10.1029/2010GB003822, 2011.