# Peer review of "The Vulcan Version 3.0 High-Resolution Fossil Fuel CO$_2$ Emissions for the United States"

_Earth System Science Data, 2019_

## Short Comment (SC1) · 29 Dec 2019

**The Vulcan Version 3.0 High-Resolution Fossil Fuel CO2 Emissions for the United States**

Kevin R. Gurney1, Jianming Liang2, Risa Patarasuk2, Yang Song2, Jianhua Huang2, Geoffrey Roest1

[revised manuscript text omitted]

| 17
18 | Table 1: Overvi
(footnotes provi | ew of data sources us
de acronym explanat | sed in generating the space/
tions). | /time-resolved Vulcan | v3.0 FFCO2 emission | ns |
|----------|-------------------------------------|----------------------------------------------|-----------------------------------------|-----------------------|---------------------|----|
|          | 0 ( )                        | F 1 1 D 1                      |                                         |                       |                     |    |

| Sector/type         | Emissions Data                              | Original spatial                  | Spatial distribution               | Temporal                   |
|---------------------|---------------------------------------------|-----------------------------------|------------------------------------|----------------------------|
|                     | Source                                      | resolution/information            |                                    | distribution               |
| Onroad              | EMFAC a CO2, EPA NEIb                       | County, road class, vehicle class | FHWA AADT                          | CCSe                       |
|                     | onroad CO 2                      |                                   |                                    |                            |
| Electricity         | CAMD f CO2, DOE/EIA g | Lat/lon, fuel type, technology    | EPA/EIA NEI Lat/Lon,               | CAMD, EIA and EPA          |
| production          | fuel, EPA NEI point CO                      |                                   | Google Earth                       |                            |
| Residential         | EPA NEI nonpoint CO                         | County, fuel type                 | FEMA HAZUS d , DOE RECS | eQUEST i model  |
| nonpoint buildings  |                                             |                                   | NE-EUI h                |                            |
| Nonroad             | NEI nonpoint CO                             | County, vehicle class             | EPA spatial surrogates             | EPA temporal               |
|                     |                                             |                                   | (vehicle class specific)           | surrogates (by SCCi)       |
| Airport             | EPA NEI point CO                            | Lat/lon, aircraft class           | Lat/Lon                            | LAWA & OPSNET k |
| Commercial          | EPA NEI nonpoint CO                         | County, fuel                      | FEMA HAZUS, DOE                    | eQUEST model               |
| nonpoint buildings  |                                             |                                   | CBECS NE-EUI I          |                            |
| Commercial point    | EPA NEI point CO                            | Lat/lon, fuel type, combustion    | EPA NEI Lat/Lon, Google            | eQUEST model               |
| sources             |                                             | technology                        | Earth                              |                            |
| Industrial point    | EPA NEI point CO                            | Lat/Lon, fuel type, combustion    | EPA NEI Lat/Lon, Google            | EPA temporal               |
| sources             |                                             | technology                        | Earth                              | surrogates (by SCC)        |
| Industrial nonpoint | EPA NEI nonpoint CO                         | County, fuel type                 | FEMA HAZUS, DOE MECS               | eQUEST model               |
| buildings           |                                             |                                   | NE-EUI m                |                            |
| Commercial          | EPA NEI nonpoint CO                         | County, fuel type, port/underway  | EPA port and shipping lane         | Flat time structure        |
| Marine Vessels      |                                             |                                   | shapefiles                         |                            |
| Railroad            | EPA NEI nonpoint CO,                        | County, fuel type, segment        | EPA NEI rail shapefile and         | Point records: EPA         |
|                     | EPA NEI point CO                            |                                   | density distribution               | temporal surrogates (by    |
|                     |                                             |                                   |                                    | SCC). Nonpoint: flat       |
|                     |                                             |                                   |                                    | time structure             |
| Cement              | Portland Cement                             | Lat/Ion                           | PCA lat/lon checked in             | Flat time structure        |
|                     | Association, USGS                           |                                   | Google Earth                       |                            |

Emissions Factors Model

- Environmental Protection Agency, National Emissions Inventory b.
- c. Federal Highway Administration, Annual Average Daily Traffic
- Federal Emergency Management Agency

[revised manuscript text omitted]

- 31 An uncertainty of  $\pm 20\%$  is applied to both the default and self-reported CO EFs regardless of fuel type. An
- 32 exception to this is for the "blast furnace gas" and "coke oven gas" fuel types in which the adjustment is  $\pm 35\%$
- 33 (Table 3). The CO EF adjustment is based on estimates of the range found in the WebFIRE database and the self-
- 34 reported CO emission factors. The CO2 EF uncertainty for coal is derived from the work of Quick (2010) while

---

## Short Comment (SC2) · 3 Feb 2020

As no one but Kevin Gurney can ever fully appreciate, the paper on Vulcan 3.0 represents a massive effort of collecting and processing huge amounts of varied data. To dig out the focused, fine-scale data relevant to CO2 emissions at 1 km by hourly resolution is a heroic effort. Yes, it requires approximations, surrogates, linear extrapolations, etc., but the Vulcan product is the gold-standard for spatial and temporal resolution at the scale of a large country. As a reviewer I can point out a few queries such as the fact that the meaning of the quotation marks in Table 2 needs clear explanation, Figure 7 does not appear to be cited in the text, and the number in line 13 on page 29 does not match what appears to be the same value in Table 6 (and the abstract) – but my substantive suggestions for the paper relate to the discussions of uncertainty. Most no-

table, the values for uncertainty throughout the text do not generally have reference to the relevant spatial and temporal scale and do not lead to summary values at the end of the paper (except for Figure 7 which is national/monthly). So Figure 10 is annual, and presumably 1 km – what kind of uncertainty are we talking about? And what kind of uncertainty are we looking at by the time we get this down to hourly? It is cool stuff.

---

## Referee Comment (RC1) · Anonymous Referee #1 · 27 Feb 2020

The manuscript publishes the 3rd version of Vulcan dataset. The dataset is very meaningful for climate change research, the method is sound.

1. What is the advantage of your dataset compared with other datasets? Please explain it in the introduction. 2. What is the improvement and update to the previous versions? On page 3, line 15-16, you only said: "we report here on improvements in methodology, resolution, uncertainty estimation, . . .". Please explain the improvements in more detail. A table summarizing the improvements and comparing different versions of the project may help readers to understand your project better. 3. You use data from numerous public dataset (page 3, line 32). Is there any inconsistent of these datasets in terms of statistical scopes and methods? Will these inconsistencies affect the uncertainty of your accounting? 4. How you calculate the uncertainty? What is the

method? 5. It seems not necessary to show territorial emissions per capita. There is a debate in the literature that territorial emission should be normalized per GDP, while consumption-based emissions that related to final consumption should be normalized per capita.

———————————————————

---

## Referee Comment (RC2) · Tomohiro Oda (Referee) · 9 Mar 2020

Dear Kevin and co-authors of the manuscript,

   This manuscript presents the updated version of the Vulcan US emission data product. The original version of the Vulcan data product (v1.?) was published as Gurney et al. (2009) in the journal, Environmental Science and Technology (EST).  Since then, the Vulcan has been used in many science and policy applications.  Thus, the significance of the Vulcan product does not need to be discussed much.  In this review, I would like to focus on some of the points where I feel more clarifications would be needed for the users of the Vulcan product.  This manuscript concisely summarizes the data and methodology used in this work, which is usually a good thing from a paper writing point of view.  However, ESSD wants stand-alone papers rather than papers that are "too" concise.  The readers will have to refer to other papers/documents and information to fully understand the data product.  Please do not hesitate to write up the data and methodology w/o concerning the text length.  Especially, emission products like this are hard to validate.  Thus, describing the data and methodology is very important to let the data users evaluate and assess the utility of the data products for their applications while identifying the limitations.  This is unusual, but essential for manuscripts to be considered for ESSD, the high-quality data publication journal.  Please further refer to Carlson and Oda (2018) ESSD for ESSD's general approach.
   Other than the above-mentioned, editorial-ish comments, I want to discuss the evaluation of the Vulcan product.  The comparison to ODIAC was certainly interesting for the fossil fuel data users.  However, as we probably both agree, such comparison does not give us much practical information as an evaluation.  Instead, what the data users, including myself, would like to see is a comparison to a previous version of Vulcan (v1.? and v2.0).  It is important to compare the updated/new product to previous versions in writing and/or analyses in order to show the improvements you claim.  A comparison to ACES (Gately and Hutyra, 2018) is also useful, given the data and methodology are very similar to many users.  Moreover, I would like to see why you think ACES is following the idea of the Vulcan.  I believe that is not very clear to many of the data users, especially the broader audience of ESSD.  I don't see the limited spatial coverage of data in ACES as a problem that prevents you from comparing two products for evaluation.  I do understand we often can't do a clean emission data comparison.  However, I still believe you could show the differences among products and explain the differences in terms of data and methodologies used.
   What this study has to do next is to compare the new/updated version of Vulcan to Hestia, given the fact that Hestia is a sort of reanalysis of the Vulcan output.  In my opinion, this should have been done in Gurney et al. (2019) ESSD or Gurney et al. (2018) JGR.  However, both papers did not cover this key important analysis.  Such comparison will provide more useful information, instead of a comparison to ODIAC, which does not actually prove any improvements regarding the new Vulcan.  A comparison to ODIAC might be okay as an analysis, but it does work as evaluation here in ESSD.  ESSD is not a place to discuss the results, but rather a platform to describe the data products and show evaluations of the products.  I also feel that how the information is transferred from the Vulcan and Hestia is unclear, and not fully documented either in Gurney et al. (2019) or this manuscript.  I would like to request more text regarding that.  I also found inaccurate version names and citations.  Version control is very important for securing the traceability of the research because the earlier Vulcan products have been heavily used.  I also thought it would be great to keep previous products for record purposes.
   That being said, I feel this manuscript needs to be significantly improved to be published as a part of ESSD.  Below, I discussed my major concerns and also listed specific line-by-line comments.  I tried not to repeat the questions and comments raised by others which have been posted on the interactive discussion.  I found all the questions and comments are very good and important.  I am very curious to hear how you respond and address them.  I hope my/our review will be useful.  It is my pleasure to

serve as a referee for this work, which will have a large impact in the research community. I look forward to receiving your response.

Sincerely,
Tomohiro Oda (tom.oda@nasa.gov)

Main concerns:

- The method/data section needs to be significantly improved. ESSD manuscripts should have a complete data table. This is the basic requirement for all the manuscripts submitted to ESSD. The data sources and journal references should be distinguished. By having such table(s), the authors can avoid having many hyperlinks in the main text and guide readers better by improving the readability. In the data table, year versions/editions of data are also important to indicate, as that could significantly change the emission estimates. Also, the base year of the geospatial data is important as it tells the data user how often the spatial data would uniquely change (or not). I have discussed this in Carlson and Oda (2018), as well as Oda (2019). Oda (2019) is my interactive review comment to the manuscript published as Janssens-Maenhout et al (2019) ESSD. You can see that I discussed and suggested the same things there.
- The data product description (data and methodology) section needs to be improved. The ESSD manuscripts need to be standalone. I feel this manuscript is not standalone. For example, the data users will need to refer to the 80-page documentation on the Gurney lab website (http://vulcan.rc.nau.edu/assets/files/Vulcan.documentation.v2.0.online.pdf) in order to understand the Vulcan product and evaluate/assess the utility of the product. I don't believe this is acceptable for ESSD. If the data users need to use additional documents and/or your guidance for the data use, that means this manuscript does not include adequate description to be published in ESSD. I don't expect the authors to include all the information documented in the 80-page documentation for the Vulcan 2.0. However, a certain (I imagine significant) amount of the data/model description needs to be added to the manuscript to fully describe how the new/updated Vulcan is developed, including how new developments are relative to the earlier versions (1.x and 2.0) of the Vulcan data product. Several of my concerns and questions in this review might be addressed by doing so.
- The version names and references do not seem to be correct. Please fix. For example, my understanding is that Gurney et al. (2009) EST describes the version 1.? of the Vulcan product. Here is what the Wikipedia page for Project Vulcan says (with versions highlighted):

  *The first Vulcan inventory (**v1.0**) was released to the public in early April 2008. **Version 1.1** was released in February 2009 and **Version 1.2** is due out in early August 2009. In addition to the data release, establishment of the Vulcan website and a press release, a video of various aspects of atmospheric transport was released on Purdue University's YouTube website and portions of the Vulcan inventory are available on Google Earth.[5] As of 2015, **version 2.2** has been published on a site hosted by Arizona State University.[6]* (source: https://en.wikipedia.org/wiki/Project_Vulcan, last access: 03/04/20).

  The documentation on the Gurney lab is for the Vulcan 2.0 (and also later?). To clearly highlight the improvements from the previous version of the Vulcan product, the background and history needs to be described a little bit more with proper version names. Again, this might not be a huge issue for other science journals, but ESSD is a high-quality data publication journal.
- Now the Vulcan data is also important as a parent data/model for the Hestia emission products (Gurney et al. 2019, ESSD), another significant emission data product the authors have developed.

The latest versions of the Hestia LA were also presented in Gurney et al. (2018) JGR (you incorrectly cited this JGR paper in the 2019 ESSD paper as a Vulcan reference, by the way). I feel that how the information is transferred from the Vulcan and Hestia is unclear, and not fully documented, either, in two previous papers or this manuscript. I believe this is fair to point out as Hestia and Vulcan are closely related and support the scientific significance of each other. W/o understanding the Vulcan product (this manuscript), the Hestia paper does not stand by itself. Understanding Hestia correctly is a key step to recognize the significance of the Vulcan product correctly.

- This manuscript lacks the evaluation of the Vulcan product. This is probably the biggest concern I have. I understand that it is challenging to fully evaluate emission products in general. However, this study has so many opportunities to evaluate the Vulcan product in terms of data, emission calculation methodologies, spatial modeling approaches and the resulting spatial emission estimates. The analyses of the per capita emissions and the Center of Mass (CoM) are good. A comparison to ODIAC is also okay, but such analyses do not really evaluate the Vulcan and prove the improvements that are brought in. We probably agree on this, but just comparing two emission fields does not provide any practical information. Instead of just presenting the ODIAC's disaggregation error, you should show the improvements you've made under the Vulcan project. Characterizing the differences from previous versions of the Vulcan product is definitely one way. I also point out that the ACES inventory (Gately and Hutyra, 2017) should provide a great opportunity to highlight the improvements you claim and place the data users in a better position to understand the new/updated Vulcan product. I don't see ACES's spatial coverage as a problem that prevents you from comparing this study to it. A comparison to ACES is a very good option to evaluate the Vulcan product, given the similarity in the data and approaches used. The authors adopted the uncertainty estimates by Gately et al. (2017). I assume the authors did so because the methodologies used in the two datasets are very similar. The authors could step back a little bit. The calculated emissions can be evaluated by comparing to other estimates. The evaluation is not just about the final gridded fields. I had a similar discussion in Oda (2019), the interactive comment to the EDGAR paper.

- A comparison to ACES is also very important as the authors need to clarify why the authors think ACES and other regional bottom-up studies are following the Vulcan's idea. It would be fair to claim it if the authors provide some explanations. The authors need to be specific in the statement about which datasets are really following the Vulcan idea. For example, the GESAPU emission project (Bun et al. 2017) started w/o following the Vulcan idea. While Bun et al. (2017) published the GESAPU grand paper in 2017 in English language, the project had started even before the publication of Gurney et al. (2009). One of the earlier studies from the GESAPU project was documented as a PhD thesis (defended in 2009). The thesis was written in Ukrainian language with an English abstract (http://ena.lp.edu.ua:8080/bitstream/ntb/3216/4/avt_01337432.pdf, K. Hamal 2009). More fundamentally, Bun et al. (2017) takes a completely different approach from the Vulcan. It is very clear by just looking at a single emission sector (Danylo et al. 2019). By the way, this is just an example to show my concern. My intention here is not to discuss who was the pioneer, but to suggest the authors to make the statement more fair.

- The description of the uncertainty calculations needs to be improved. It is unclear to me how the uncertainty estimates are made. This is clearly related to the fact that the data and methodologies are not fully described. It is okay to cite other uncertainty estimates, but the lack of the details of the Vulcan methodologies and the uncertainty calculations makes it hard to assess if the uncertainty assessment was appropriately done in a scientifically satisfying fashion. Have you bumped up the uncertainties for the years where you used scaled values? I assume you did not include the uncertainties associated with the spatial modeling. Is that correct? The uncertainty estimates provided are model/inventory specific as FFDAS (Rayner et al. 2010; Asefi-Najafabady et

al. 2014) does.  Given that, is it fair to say the uncertainty estimates are likely to be underestimated?  This seems to be partially addressed by the top-down work by Basu et al. (2019) though.

- Another important thing to point out here for ESSD is the data description section. The section needs to accurately describe the data you provide under the DOI the authors indicated.   In the main text, the authors mentioned the Vulcan is provided at the 1x1km resolution, as done in ODIAC.  But that statement does not seem to be accurate.  I see the Vulcan 3.0 data provided in two data formats.  One Vulcan emission field is provided on a lambert conformal projection grid and the other one is on a regular latitude-longitude grid.  Which one is the native emission field (one with lambert-conformal projection)?  Do the authors plan to provide the Vulcan 3.0 in two grid formats?  I believe the earlier versions of the Vulcan data were provided in two grid formats.  Currently, we only see the data product provided on the lambert-conformal projection.  My other concern here is about version control.  Especially, the authors should describe how to deal with the version control for both Vulcan and Hestia consistently.  Hestia already has several versions released.  Are all the existing versions of Hestia based on the Vulcan 3.0 output?  This should be explained to inform the readers how the Vulcan update impacts on Hestia updates.  Lastly, the statement regarding the Vulcan update seems to be very optimistic, observing the history of the Vulcan and ACES updates.  Given the labor-intensive approach, it does not seem to be feasible to fully update the Vulcan data product as frequently as claimed.  I would expect that the authors do need some sort of heavy simplification to make it happen.

Line by line comments:

Abstract
P1, L7: I feel the current manuscript does not provide the complete description of the Vulcan 3.0.
P1, L9: It is unclear to me how these uncertainties are calculated.  This is also asked by someone at the interactive discussion.  Please clarify.  Also, how should we be using these uncertainty estimates, especially ones at the grid level?  It would be good to have clear guidance from the authors in order to avoid misuse by the data users.  By the way, I assume these uncertainty estimates are inventory/model-specific uncertainties, like ones provided by FFDAS.
P1, L10: Are these per capita CO2 estimates significantly different from what you would get from the state level CO2 emission estimates?  If so, why?  I believe this is another good way to evaluate the Vulcan 3.0 and allow the authors to characterize the differences from other estimates.
P1, L14: The total emissions from ODIAC are based on the CDIAC estimates (Oda et al. 2018).  When the total emissions are examined, the authors should discuss using the CDIAC estimates (no impact from emission disaggregation).  The data users would be curious to know emission differences due to the differences in the emission calculation methods (Vulcan vs. CDIAC).  This is the place the authors can highlight their own unique calculation methods and the improvements/advancements in relation to previous versions of the Vulcan product or existing other estimates.
P1, L17: How are Vulcan 1x1km emission fields at cities different from Hestia emission fields?  Do we still need Hestia?  The language here is pretty similar to what was claimed for Hestia.  What is the significance of Hestia now when the Vulcan 3.0 is available to us?  Is it just a matter of the spatial resolutions (multi-scale vs. gridded)?  I am asking this because the differences between the Vulcan and Hestia fields are not well explained in either Gurney et al. (2019) or this manuscript, thus unclear.  I believe comparisons (verbally and quantitatively) of Hestia and Vulcan will clear this up.

Introduction

P2, L5: Citing Durant et al. (2011) and Bellassen et al. (2015) seems to be a little stretch, given the sentence the authors wanted to support.  Also, Janssens-Maenhout et al. (2013) is a book review of a National Research Council (NRC) report.  I do not have access to this book review, so I am curious to learn if there was any particular reason that this needs to be cited, instead of the original report by Pacala et al.  Citing the original Pacala report seems to be more sensible.

P2, L8: How about citing studies like Callender (1938) and Keeling (1973) here?

P2, L9: Such as?  The authors should not assume the broader audience of ESSD are familiar with fossil fuel CO2 big names, such as CDIAC.

P2, L13: The link to the Marland et al. paper seems to be no longer active.  Please update.

P2, L15: Liu et al. (forward modeling), Yadav et al. (inversion framework), and Gaubert et al. (inversion inter-comparison).  Given the statement the authors wanted to support, the authors could choose more appropriate studies that actually implement inversions with observational data.

P2, L17: and combinations of two (e.g. Rayner, Ghosh, and Ou).  The authors should also add Asefi-Najafabady et al. (2014) here as the study also creates a combination of the population and nightlight in the modeling framework.

P2, L19: This needs to be fixed.  Wang et al. (2013) used nightlight data only for small fraction of the total emissions.  The majority of diffused emissions are distributed using population data.  I would propose to include EDGAR here.  EDGAR's inclusion of a variety of geospatial information was pioneering and has uniquely defined the EDGAR data product.

P2, L22: Asefi- -> Asefi-Najafabody.. Why isn't Rayner et al. (2010) mentioned here?  The two studies both do, in different ways.

P2, L25: Ivanova et al. is about carbon footprint and it does not present subnational distributions (only country scale).  How about organizing these studies by region (e.g. A and B for Europe, C and D for China…)?

P2, L27: I don't not disagree with this, but I am not sure if we can generalize this like that.  For example, REAS (Ohara et al. 2007; Kurokawa et al. 2013) did have CO2 estimates together with air pollutant emissions.  I feel the authors need to carefully discuss this, unless the authors are very sure of the histories of the studies cited.

P2, L28: O'Hara -> Ohara (Japanese surname)

P2, L30: The authors should be careful about this statement.  The authors need to be specific which one(s) are following the Vulcan project by explaining why.  I feel it is unfair to claim this w/o doing so.  We should not just judge that by looking at the publication year.  I am sure and I agree that the Vulcan project has inspired many emission studies though.

Methods

P3, L3-: I am a little bit confused with this paragraph.  I imagine the authors wanted to state the Vulcan only indicates the scope 1 emissions.  Correct?  Maybe the authors also wanted to emphasize the limitations of Vulcan?  Do the authors plan to have Scope 2 Vulcan?  What was the intention here?  I agree with this paragraph here, but I would like to suggest to improve the flow.

P3, L15: The improvements are not highlighted while the authors could do many things to do so (see my comments earlier).

P3, L19: The readers would be curious to know what is the utility of Vulcan relative to Hestia at urban domains.

P3, L32: Gurney et al. 2019 -> Gurney et al. 2019a?

P3, L34: Due to the lack of details, it is difficult to realize the actual native resolution of the emissions. Also, with the word "native", it is more difficult to clearly distinguish Hestia and Vulcan in terms of the basic approach. I believe this will be addressed by improving the data/method section.

P4, L2: Did you bump up the uncertainty to account for these "conditioning"?

P4, L6: And height for aviation emissions?

P4, L10: This is not acceptable for ESSD. ESSD wants a standalone paper.

P4, L11: Incorrect citation. Gurney et al. 2009 is for version 1.x. There is no peer review papers for the Vulcan 2.0, I believe. An extended Vulcan overview needs to be added in this manuscript in order to adequately educate the data users.

P4, L13: This does not seem to be consistent with the data product distributed under the DOI indicated in the manuscript.

P4, L15: It is not clear to me what was done for uncertainty calculations and what is missing.

P4, L17: Table 1 needs to be improved. The data sources and data citations need to be distinguished. Also see my comment earlier.

P8, L18: This is an emission disaggregation. Would you be able to evaluate the disaggregation error by comparing the Vulcan fields to Hestia?

P8, L29: Does this remain the same over the 6 years?

P11, L12: How did the authors define the representative points of the sources (e.g. center of the facility or multiple outlet/smoke stacks)?

P12, L5: Is this a mean climatology? If so, the number of data points and standard deviation (SD = proxy for uncertainty?) would be of interest to the readers.

P12, L11: Does this mean the methodology is essentially the same in this study and Gately et al.?

P12, L19: How many of them?

P18, L2: How well does NONROAD model work? Any evaluation? These uncertainties should be aliasing to the final Vulcan uncertainties (the same comment for other models that are used internally). I would not worry too much if the model output was used for downscaling (proxy approach). However, the model performance here has impact on the accuracy of the bottom-up emission estimates.

P20, L4: Figure 2. SD for the lines?

P22, L23: Gurney et al. (2009) is an incorrect citation for the version 2.0 Vulcan.

P23, L8: This is about emission estimations. How about emission spatial locations/distribution? How often do the underlying data change? This is a very important thing to know when reconciling the differences among spatial emission fields (bottom-up emission estimates & the spatial distributions) from different products.

Results

P23, L30: Why year 2012? The multi-year emissions are achieved using the 2011 as a base year.

P23, L31: Uncertainty range in %?

P24, L30-: The total US emissions from Vulcan can be compared to the other available estimates. Such comparisons should allow the authors to calculate a meaningful emission difference and provide opportunities for discussions.

P24, L 7: These kinds of analyses are interesting. However, the first thing the authors should have done was to evaluate the performance of the Vulcan. Validating emission products is challenging and often can't be done cleanly. I share the difficulty. However, in the case of this study, the authors could do very meaningful evaluations. For example, the authors can do the same analyses using other estimates (e.g. reported values and data products) and compare that to numbers from the Vulcan. Since the Vulcan has its own unique emission calculations, we would like to know the accuracy of the emission estimates. At least the authors should compare those numbers at

aggregated levels to show the validity of the Vulcan emission calculation approaches and characterize the differences from other estimates, from data and methodology perspectives.

P26, L4: Figure 4. The same comment as above. Is this very different from what we would get from other estimates? This isn't an evaluation, but application. We would like to see evaluations first.

P27, L3: Figure 3 (b). The authors could have roughly compared this aggregated 0.1 degree Vulcan field to the earlier version of Vulcan to quantify the differences (= improvements). The authors could do the CoM analysis for both previous and new Vulcan data by scaling total? The differences in the trajectory of the CoM from different versions of the Vulcan should be a great tool for evaluation.

P28, L3: Figure 6. It would be helpful to have a scale here.

P28, L8: how different is this from other exiting monthly estimates (e.g. CDIAC and EDGAR, ACES)?

P29, L3: At least, Bun et al. (2019) is probably not following the Vulcan idea.

P29, L5: See one of my major concerns.

P29, L8: Which ODIAC? Please indicate the data version (ODIACYYYY, YYYY=year version) and include the data citation.

P29, L8: ODIAC are not on the same 1km grid.

P29, L11: If the authors went back to the CDIAC estimate (input data for ODIAC), the authors could have done a cleaner comparison for the total emissions.

P29, L26: Liang et al. (2019) is not in the reference. Probably you meant Gurney et al. (2018) JGR.

P30, L15: Gurney et al. (2018) reported the domain wide difference for LA as -1.5%. How can we reconcile the result from Gurney et al. (2018) and the statement here?

P31, L4: This also needed to be done using previous versions of the Vulcan and the Hestia. It should have provided a sort of measures of the improvement brought in and highlighted the significance of the Vulcan or Hestia.

P32, L1: Figure9: OK, so the authors seem to have the lat/lon regular grid version of the Vulcan product. Do the authors plan to provide that version of the data product? It seems that the regular grid version of the Vulcan is not covered in the data section.

P33, L1: This is great to learn. However, as mentioned earlier, there are many comparisons to show the validity and/or improvements of the Vulcan 3.0, other than the top-down estimation.

P33, L5: What would be the difference between the updated and pervious Vulcan data products when aggregated to 10km? This is now really meaningful as the Vulcan serves as a parent model for the Hestia. And then my question would be what are the differences between Hestia and Vulcan at 1x1km in city domains?

P32, L3: Figure 10. I feel the color scale do not well highlight the important low emissions around cities. As the authors claimed (also Oda et al. (2019) reported), the urban-rural transitioning areas indicate larger errors. Please improve the color scale to show emission distributions over the transition areas well.

P34, L6: How about doing the same analysis using earlier versions of the Vulcan data to highlight the differences/improvements?

P34, L16- : Data section needs to be improved. I feel this section does not adequately explain the data distributed under the DOI. See Carlson and Oda (2018) and/or Oda (2019) for what level of information ESSD wants the authors to provide. Do you plan to distribute the lat/lon regular grid version of the data, too? How about the per cap emission map data?

P34, L24: Looking at the history of bottom-up inventories (Vulcan and ACES), it seems to be extremely challenging to make the data update happen as frequently as claimed here. I would be more convinced if the authors describe how the authors plan to do the update. Please be clear about what will be updated and what remain the same in the update process. For example, if the authors just scale the total emissions by keeping the other things the same, such (partial) update

is totally doable.  However, the significance as bottom-up emissions would be significantly lowered.

Conclusion
Please make necessary changes to be consistent with the contents presented in the sections earlier.

Supplementary Information
P1, Table S1: Please characterize the differences you observed here.  What is driving them?  Numbers are enormously large for some states.
P2, Table S2: Please define the aggregate type and sub-type code.
P3, Table S3: Please define the eQUEST building code.
P3, Table S4: Please defined those terms.  Also, please provide justification to the level where at least we can feel those are reasonable.
P5, Figure S1: Please add the total number of the VMT.
P5, Table S5: The same comment as above.  Did the authors bump up the uncertainty for the gap-filled ones?

References cited in this review
- Basu et al. (2019) as cited in the main text
- Bun et al. (2017) as cited in the main text
- Callender (1938) https://rmets.onlinelibrary.wiley.com/doi/abs/10.1002/qj.49706427503
- Carlson and Oda (2018) ESSD https://www.earth-syst-sci-data.net/10/2275/2018/
- Danylo et al. (2019) https://link.springer.com/article/10.1007%2Fs11027-019-9846-z
- Gately and Hutyra (2018) JGR as cited in the main text
- Gurney et al. (2009) EST as cited in the main text
- Gurney et al. (2018) JGR as cited in the main text
- Gurney et al. (2019) ESSD as cited in the main text
- Keeling (1973) https://www.tandfonline.com/doi/abs/10.3402/tellusa.v25i2.9652
- Oda and Maksyutov (2011) ACP as cited in the main text
- Oda et al. (2018) as cited in the main text
- Oda (2019) Interactive comment on "EDGAR v4.3.2 Global Atlas of the three major Greenhouse Gas Emissions for the period 1970–2012" by Greet Janssens-Maenhout et al., https://www.earth-syst-sci-data-discuss.net/essd-2018-164/essd-2018-164-RC2.pdf
- Oda et al. (2019) https://link.springer.com/article/10.1007/s11027-019-09877-2
- Rayner et al. (2010) JGR as cited in the main text

---

## Short Comment (SC3) · 11 Mar 2020

Additional comments on ESSD-2019-154 (Vulcan US emissions)

I appreciate that our research community focuses on emissions products. Definitely we need those efforts, globally and nationally! I also like that ESSD plays a helpful role in promoting, certifying and sharing those products.

Unfortunately, for several reasons, this most-recent Vulcan product as submitted and as described fails to meet many requirements and expectations for ESSD. In several comments below I echo and emphasize points made by reviewer #2. As chief editor for ESSD for more than 10 years, I hope I offer useful and well-informed viewpoint.

Note: I read both this paper and the prior (Hestia) publication in ESSD.

Overall, I find very little about data, methods, validation, etc. to give a reader / user confidence. I find the registration requirement at ORNL unacceptable. I repeat, in the strongest terms, the recommendation of reviewer #2: read the guidelines!!! (https://www.earth-syst-sci-data.net/10/2275/2018/)

Reader has no idea of sequence or versions. Hestia based on Vulcan? Vulcan builds on lessons learned from Hestia? This version, apparently Vulcan 3.0, improves on which prior version(s)? What improvements? How do the authors confirm improvements? Does the reader need to go back to Nature papers in 2002?

What external sources? How accessible? How reliable? Near the end the authors write "depending upon support and the availability of data sources described in this study". Will availability prove problematic? For all sources? Specific sources?

Manuscript needs two tables. First, a clear sequence of prior and related products leading to Vulcan 3. Do not make readers guess or search. Provide reliable up-to-date links. If not open access, make them (all) open access. Second, a clear comprehensive list, in table form, of all sources. Perhaps 50 or more, no matter. Let readers know what you used, what version you started from, with active certified links to all sources. Most users do not want to try to follow every step, but authors must nonetheless provide exact guidance and source information.

This reader / editor does not like the "Vulcan'Science'Methods'Documentation,'Version'2.0" It gives no information about date or version. In too many cases it appears to derive from 2002. That .pdf has undergone no review, no critical reading, etc.; it looks like a lab report. Users will not find it useful or reliable. I also read at least partial overlap between manuscript and lab report. If useful, put it all in the manuscript.

Speaking of Hestia, I find only one mention accompanied by a single citation. Why, in Figure 10, do they not show an LA example? No discussion, intercomparison, etc. Meanwhile, Hestia clearly specifies "Hestia-LA data product are supplied by output of the Vulcan Project" (https://doi.org/10.5194/essd-11-1309-2019) Do the author intend a series of Hestia-cityname products, apparently isolated from Vulcan manuscripts? Also apparently substantial text overlaps between that document and this? What did our similarity test show?

I find efforts to compare with ODIAC unsatisfactory. What does figure 9 tell me, quantitatively and reliably? Nothing. Because authors have not provided confidence trail for development of Vulcan 3, a snapshot comparison to ODIAC proves meaningless. What product works well for what purposes, with what uncertainties, and why? Can we assign differences to night light data in one but not the other? By how much would that impact? Which product needs what improvements? As a data journal, ESSD needs to ensure readers can ask and answer such questions. How does this work contribute to that discussion?

I find too many proprietary tools and legend errors. Figure 5: Copyright Ó National Geographic Society. Figure 6 (where authors have reversed the line colors: red designates months (not years) while purple designates years (not months): Copyright © Esri. Figure 10: ArcMapTM by Esri using the World Imagery basemap layer (Copyright © Esri). Proprietary copyrighted tools and sources are not acceptable in ESSD!

Other reviewers have pointed out many technical errors. I find the entire manuscript unreliable with key information hidden or unaccessible.

With many months of effort and persistent cooperation by authors and reviewers, ESSD helped those authors bring the EDGAR product into a successful published product. I believe ESSD can and should do the same here, with the first step being to recognize and acknowledge current substantial deficiencies. ESSD = open access. Vulcan = not (yet).

---

## Author Comment (AC1) · 13 Mar 2020

Additional comments on ESSD-2019-154 (Vulcan US emissions)

I appreciate that our research community focuses on emissions products. Definitely we need those efforts, globally and nationally! I also like that ESSD plays a helpful role in promoting, certifying and sharing those products.

Unfortunately, for several reasons, this most-recent Vulcan product as submitted and as described fails to meet many requirements and expectations for ESSD. In several comments below I echo and emphasize points made by reviewer #2. As chief editor for ESSD for more than 10 years, I hope I offer useful and well-informed viewpoint.

Note: I read both this paper and the prior (Hestia) publication in ESSD.

Overall, I find very little about data, methods, validation, etc. to give a reader / user confidence. I find the registration requirement at ORNL unacceptable. I repeat, in the strongest terms, the recommendation of reviewer #2: read the guidelines!!! (https://www.earth-syst-sci-data.net/10/2275/2018/)

We will attempt to respond to specific comments regarding the data, methods and validation in order to increase user confidence.

Extensive discussion with both ORNL and the editor with whom we worked at ESSD led us to believe that the ORNL DAAC had availability and access rules consistent with those expected and outlined in the guidelines - which we did read (ps. the link provided by the reviewer does not land on a page related to guidelines, worth noting when communicating with authors in the future). We followed:

"Copernicus Publications requests that all ESSD authors deposit their data corresponding to journal articles in reliable (public) data repositories, assign digital object identifiers, and properly cite data sets as individual contributions. Please find your appropriate data repository in the list of data centres supporting the ESSD criteria or consult with ESSD editors."

Many weeks were devoted to discussion on this topic. We thought we had reached agreement on the issue of access. I just email the ORNL DAAC and they do require a sign-in. This is the text of the email they sent in response to my asking about the sign-in producedure:

"Access to all NASA Earth Science data requires a user registration through Earthdata Login. There have been multiple discussions with multiple journals about this in the past, including Kirsten Elger (ESSD Editor in Chief). Apart from requiring that login, there is no restriction on access or use of the data. The information a user provides in the registration process is only used in aggregate to understand the distribution of users.

I've spoken with Kirsten in the past, including at the recent AGU meeting. I suggest the editor discuss the matter with her and one or both of them can reach out to me if my answers and past discussions with Kirsten do not resolve the matter.

See this text on the Earthdata Login page (https://urs.earthdata.nasa.gov/).

Why must I register?

The Earthdata Login provides a single mechanism for user registration and profile management for all EOSDIS system components (DAACs, Tools, Services). Your Earthdata login also helps the EOSDIS program better understand the usage of EOSDIS services to improve user experience through customization of tools and improvement of services. EOSDIS data are openly available to all and free of charge except where governed by international agreements.

This is not something where we have control. This is defined at the NASA Earth Sciences Division level and applies to all NASA Earth Science Data."

Please understand my frustration at having spent weeks discussing this topic with the assigned editor at ESSD and the ORNL DAAC upon which we assumed we had agreement only to find that there is not agreement. If a sign-in remains a barrier we will have to withdraw our paper and submit elsewhere. This DAAC is strongly encouraged by our funders as this project is under the Carbon Monitoring System umbrella.

Reader has no idea of sequence or versions. Hestia based on Vulcan? Vulcan builds on lessons learned from Hestia? This version, apparently Vulcan 3.0, improves on which prior version(s)? What improvements? How do the authors confirm improvements? Does the reader need to go back to Nature papers in 2002?

We are not clear on what relevance Hestia, a separate project, has to do with the current manuscript? Hestia is mentioned once in the current manuscript as follows:

"Corrections to location information were made in urban domains associated with the Hestia Project : the Los Angeles Basin, Baltimore, Salt Lake City, and Indianapolis (e.g. Gurney et al., 2018; 2019b)."

This was only to point out that location information was corrected in the four isolated geographic locations through the Hestia project and the relevant citations made. Hestia is not based on Vulcan – nowhere in the manuscript has any relationships been suggested other than these location corrections. We do not understand why a discussion of Hestia is necessary for this manuscript?

The Nature 2002 paper has nothing to do with Vulcan, Hestia, or inventories at all. It is a paper on atmospheric $CO_2$ inversions at the global scale, included to offer the reader a foundational paper on inversions and how they use bottom-up gridded emissions information. The reviewer must have a fundamental misunderstanding.

The need to notate and or otherwise reference previous versions of Vulcan is difficult: at the time of its publication (2009), standardized data journals such as ESSD were less common and standard data practices were not as common as they are now. Hence, it was published in a journal that did not have the requirements current data journals have. The documentation was available but not in the same way or form we are attempting to do so presently. Hence, there is no standardized pedigree to point to. A solution is to write the Vulcan version 3.0 as standalone and put all documentation into this paper such that every detail is included. Assuming we proceed with ESSD, we will confer with the reviewer or contact editors on this topic.

What external sources? How accessible? How reliable? Near the end the authors write "depending upon support and the availability of data sources described in this study". Will availability prove problematic? For all sources? Specific sources?

We are not entirely clear what the reviewer is referring to with regards to the questions "What external sources? How accessible? How reliable?" We take this as a general query regarding the input data used to generate the Vulcan estimate. Regarding the "external sources" - the data sources are provided in the manuscript – all of which are available through the citations and URLs provided. All input data is identified – there are no other sources.

Regarding the quoted statement - this was included because, as was noted in the manuscript text, the input data to the Vulcan data product is primarily "regulatory" data (all publicly available). The timing of the release of these input data varies with considerable latency. Furthermore, the data structures and data fields have varied in the past and hence, cannot be considered absolutely guaranteed in form indefinitely. The accessibility and reliability are difficult for us to comment on – US government agencies can change data policies or alter collection procedures. Hence, we felt that this caveat was important to include so as the reader is aware that continued temporal extension (beyond the current 2010-2015 time period) or version updates cannot be guaranteed in the future (the current version 3.0 can be, however). The allusion to support also make clear that the magnitude of this work cannot be extended or improved without continued support from funding agencies.

It may be best to remove this statement as it would be difficult to ascertain which data sources have guaranteed availability into the future versus others.

Manuscript needs two tables. First, a clear sequence of prior and related products leading to Vulcan 3. Do not make readers guess or search. Provide reliable up-to-date links. If not open access, make them (all) open access. Second, a clear comprehensive list, in table form, of all sources. Perhaps 50 or more, no matter. Let readers know what you used, what version you started from, with active certified links to all sources. Most users do not want to try to follow every step, but authors must nonetheless provide exact guidance and source information.

We will add the requested information to the manuscript and hope it satisfies the needs of the reader. As with previous comments here, we will consider writing Vulcan 3.0 as a standalone data product in which all information regarding its construction and inputs are presented here avoiding any reference to previous versions.

This reader / editor does not like the "Vulcan'Science'Methods'Documentation,'Version'2.0" It gives no information about date or version. In too many cases it appears to derive from 2002. That .pdf has undergone no review, no critical reading, etc.; it looks like a lab report. Users will not find it useful or reliable. I also read at least partial overlap between manuscript and lab report. If useful, put it all in the manuscript.

We will remove reference to the Vulcan 2.0 documentation. We have added additional relevant details from the earlier documentation to the manuscript. Our aim is for the Vulcan 3.0 to stand alone as a data product without reference to the previous version.

All links are up-to-date. It is hard to establish their reliability as they are data supplied by regulatory/government agencies. They are active and come with no barriers of any kind to their free and open use. They will be double-checked.

Speaking of Hestia, I find only one mention accompanied by a single citation. Why, in Figure 10, do they not show an LA example? No discussion, intercomparison, etc. Meanwhile, Hestia clearly specifies "Hestia-LA data product are supplied by output of the Vulcan Project" (https://doi.org/10.5194/essd-11-1309-2019) Do the author intend a series of Hestia-cityname products, apparently isolated from Vulcan manuscripts? Also apparently substantial text overlaps between that document and this? What did our similarity test show?

The Hestia Project is a different effort and does not constitute any inputs to the Vulcan system. Hestia is mentioned once in the manuscript with two (not one) citations. There was no particular reason to show the Los Angeles domain in figure 10. Hestia performs additional space/time conditioning beyond what the Vulcan Project produces. They are numerically consistent at aggregate spatial scales (which varies by sector) hence, a comparison between Vulcan and Hestia is of marginal value. We will pursue this in the future but we do not see why Hestia is relevant to this manuscript or why a comparison between Hestia and Vulcan would be considered relevant.

We cannot find the quoted text in this manuscript so it is difficult to respond to aspects of the question other than to not that Hestia can be thought of as a nested effort within Vulcan – it zooms to finer detail with Vulcan as constraint at the aggregate scale. The Hestia project will continue and may add cities. They will use the Vulcan output and cite the Vulcan version 3.0 paper (wherever it is published) as the input. They will be written separately as they employ methods relevant to Hestia. The overlap in text on the LA paper is merely to provide context within the Hestia papers – sensible given that they derive much from the Vulcan effort. However, the reverse does not occur.

I find efforts to compare with ODIAC unsatisfactory. What does figure 9 tell me, quantitatively and reliably? Nothing. Because authors have not provided confidence trail for development of Vulcan 3, a snapshot comparison to ODIAC proves meaningless. What product works well for what purposes, with what uncertainties, and why? Can we assign differences to night light data in one but not the other? By how much would that impact? Which product needs what improvements? As a data journal, ESSD needs to ensure readers can ask and answer such questions. How does this work contribute to that discussion?

We respectfully disagree. Fossil fuel $CO_2$ emissions have no reference for validation – ODIAC does not (nor did it present such documentation in the ODIAC ESSD paper) nor do ACES, EDGAR, FFDAS, CDIAC etc. Hence, comparisons offer the only opportunity to better understand the respective data products, something difficult to do in isolation where there is no validation mechanism (yet). As we stated in the manuscript:

"we perform comparison to the ODIAC output over the Vulcan domain in the hope of providing insight into one or both of the emission estimates"

The lower threshold cutoff is evident in both figure 8 and figure 9 and is an important distinction that is driven by the use of nightlights in ODIAC. There are four paragraphs on the comparison with difference metrics. Figure 9 shows that there is a wider range of per gridcell emissions in Vulcan compared to ODIAC with greater magnitude differences at the lower-emitting gridcell end of the range (likely due to the low-end threshold cutoff in nightlights). The differences are relevant as they point to the very different spatial content of a bottom-up versus a downscale or proxy distribution approach, a distinction that the community must consider as they use these data products in various contexts and applications.

I find too many proprietary tools and legend errors. Figure 5: Copyright Ó National Geographic Society. Figure 6 (where authors have reversed the line colors: red designates months (not years) while purple designates years (not months): Copyright © Esri. Figure 10: ArcMapTM by Esri using the World Imagery basemap layer (Copyright © Esri). Proprietary copyrighted tools and sources are not acceptable in ESSD!

The one legend error in Figure 6 will be corrected.

As for the reference to tools in three of the 10 figures in the manuscript we had iterated with the editor on this topic as well and thought the final drafting was acceptable. Indeed, we initially had not included that copyright attribution on these figures but were asked to include them. The copy right only refers to the "basemap" NOT the data overlaid on the basemap (the Vulcan emissions are the data in these cases). In examining other papers in ESSD, basemaps are found throughout without copyright attribution. Hence, perhaps there is misunderstanding or confusion regarding the requirements?

Other reviewers have pointed out many technical errors. I find the entire manuscript unreliable with key information hidden or unaccessible.

We are responding to the comments made by other reviewers in separate response files.

We do not know how to respond to the general comment of unreliability, hidden/unaccessible informrmation other than to attend to specific comments along these lines and review the manuscript for missing details.

With many months of effort and persistent cooperation by authors and reviewers, ESSD helped those authors bring the EDGAR product into a successful published product. I believe ESSD can and should do the same here, with the first step being to recognize and acknowledge current substantial deficiencies. ESSD = open access. Vulcan = not (yet).

We can only assume the last comment refers to the issues at the ORNL DAAC? We have commented previously on this.

---

## Author Comment (AC2) · 1 Apr 2020

As no one but Kevin Gurney can ever fully appreciate, the paper on Vulcan 3.0 represents a massive effort of collecting and processing huge amounts of varied data. To dig out the focused, fine-scale data relevant to CO2 emissions at 1 km by hourly resolution is a heroic effort. Yes, it requires approximations, surrogates, linear extrapolations, etc., but the Vulcan product is the gold-standard for spatial and temporal resolution at the scale of a large country.

As a reviewer I can point out a few queries such as the fact that the meaning of the quotation marks in Table 2 needs clear explanation

These were symbols to represent a repeat of a value. However, this was not the best choice, so we have been explicit in the repetition of values in the table.

Figure 7 does not appear to be cited in the text

This has now been referenced on page 27, line 18 at the end of the sentence as "...... in the months of July and August."

the number in line 13 on page 29 does not match what appears to be the same value in Table 6 (and the abstract)

They are not intended to be the same. On page 29, line 13 the value is the result of using a mask for ODIAC. This is slightly different from the raw Vulcan result (which is presented in Table 6). We have added text to clarify this as follows:

"The same mask applied to Vulcan results in $FFCO_2$ emissions of 1553.8 TgC/yr (distinct from the unmasked Vulcan total of 1589.9 TgC/yr) or a difference of 100.3 MtC/yr (7.6%)."

but my substantive suggestions for the paper relate to the discussions of uncertainty. Most notable, the values for uncertainty throughout the text do not generally have reference to the relevant spatial and temporal scale and do not lead to summary values at the end of the paper (except for Figure 7 which is national/monthly). So Figure 10 is annual, and presumably 1 km – what kind of uncertainty are we talking about?

We have now included the 95% CI boundaries in Table 6 for the totals in each year. All totals in the text now have the 95% CI boundaries included. Uncertainty for Figure 10 is now provided in caption.

And what kind of uncertainty are we looking at by the time we get this down to hourly?

There is no distinct uncertainty applied at the hourly versus annual scales. We have added text to uncertainty sections to make it clear how this uncertainty would be represented at different scales.

It is cool stuff.

---

## Author Comment (AC3) · 1 Apr 2020

The manuscript publishes the 3rd version of Vulcan dataset. The dataset is very meaningful for climate change research, the method is sound.

1. What is the advantage of your dataset compared with other datasets? Please explain it in the introduction.

We have added text in the introduction that directly responds to this query. Specifically we say "Vulcan is distinct from ODIAC in that it includes detail regarding combustion sector, combustion sub-sector (e.g. by vehicle class, building type), combustion process (e.g. boiler, turbine, engine), and a detailed fuel characterization (e.g. individual petroleum fuels, coal grade). Though reported here as gridded output, the underlying emissions content is quantified as individual point, line, and polygon source elements and as such, is distinct in potentially providing finer resolution in the future. Finally, unlike top-down inventories, typically produced at the global scale, Vulcan is constructed from the bottom-up, relying less on indirect spatial proxies (e.g. nighttime lights) and more on detailed mapping of physical entities such as roadways and factories."

2. What is the improvement and update to the previous versions? On page 3, line 15-16, you only said: "we report here on improvements in methodology, resolution, uncertainty estimation, . . .". Please explain the improvements in more detail. A table summarizing the improvements and comparing different versions of the project may help readers to understand your project better.

We are altering the manuscript to no longer make reference to previous version but write this as a standalone paper. Tracing the improvements and methods in previous versions is a large and potentially impossible task. Vulcan version 3.0 will stand alone and the manuscript will include all information related to its methods and results.

3. You use data from numerous public dataset (page 3, line 32). Is there any inconsistent of these datasets in terms of statistical scopes and methods? Will these inconsistencies affect the uncertainty of your accounting?

We are not entirely sure what is meant by this question. The data do come from many different sources and they will most often have differing levels of reliability, uncertainty, consistency, and so on. The will be imparted to the results in this manuscript, for sure, and we attempt to reflect that in the uncertainty analysis, but it is probably a simplification of the true uncertainty. The true uncertainty is challenging to quantify given the limitations in input data documentation and/or procedures. It must be noted that unlike other data products of a similar kind published in ESSD, Vulcan uses a wide array of regulatory data, which typically does not include careful documentation, release versioning, uncertainty estimation, etc. This makes it very difficult to have the same procedural documentation as might be found in global granular emissions efforts.

4. How you calculate the uncertainty? What is the method?

Perhaps it was missed but we include a section on uncertainty within each of the sub-sections to section "2.1 Data and processing". We have included more detail in each of these given the comments of other reviewers.

5. It seems not necessary to show territorial emissions per capita. There is a debate in the literature that territorial emission should be normalized per GDP, while consumption-based emissions that related to final consumption should be normalized per capita.

We generally agree that the choice of normalization is a critical conditioner to interpretation. We were not trying to make a particular policy or social science point when normalizing to population. Normalizing by population was done to offer a constrast to absolute magnitudes and emphasize why population-normalized values vary according to sector. A normalization by GDP is done and now included in the SI – it shows a very similar result.